# Prebiotic Xylo-Oligosaccharides Ameliorate High-Fat-Diet-Induced Hepatic Steatosis in Rats

**DOI:** 10.3390/nu12113225

**Published:** 2020-10-22

**Authors:** Sanna Lensu, Raghunath Pariyani, Elina Mäkinen, Baoru Yang, Wisam Saleem, Eveliina Munukka, Maarit Lehti, Anastasiia Driuchina, Jere Lindén, Marja Tiirola, Leo Lahti, Satu Pekkala

**Affiliations:** 1Faculty of Sport and Health Sciences, University of Jyväskylä, FI-40014 Jyväskylä, Finland; sanna.t.k.lensu@jyu.fi (S.L.); elina.e.makinen@jyu.fi (E.M.); maarit.t.lehti@jyu.fi (M.L.); anastasiia.a.driuchina@jyu.fi (A.D.); 2Food Chemistry and Food Development, Department of Biochemistry, University of Turku, FI-20014 Turku, Finland; raghunath.pariyani@utu.fi (R.P.); bayang@utu.fi (B.Y.); 3Department of Future Technologies, University of Turku, FI-20014 Turku, Finland; wisam.tariqsaleem@utu.fi (W.S.); leo.lahti@utu.fi (L.L.); 4Institute of Biomedicine, University of Turku, FI-20014 Turku, Finland; laevmu@utu.fi; 5Department of Clinical Microbiology, Turku University Hospital, FI-20521 Turku, Finland; 6Veterinary Pathology and Parasitology, University of Helsinki, FIN-00014 Helsinki, Finland; jere.linden@helsinki.fi; 7Department of Environmental and Biological Sciences, University of Jyväskylä, FI-40014 Jyväskylä, Finland; marja.tiirola@jyu.fi

**Keywords:** prebiotic, oligosaccharides, gut microbiota, fatty liver, metabolism, mitochondria

## Abstract

Understanding the importance of the gut microbiota (GM) in non-alcoholic fatty liver disease (NAFLD) has raised the hope for therapeutic microbes. We have shown that high hepatic fat content associated with low abundance of *Faecalibacterium prausnitzii* in humans and, further, the administration of *F. prausnitzii* prevented NAFLD in mice. Here, we aimed at targeting *F. prausnitzii* by prebiotic xylo-oligosaccharides (XOS) to treat NAFLD. First, the effect of XOS on *F. prausnitzii* growth was assessed in vitro. Then, XOS was supplemented or not with high (HFD, 60% of energy from fat) or low (LFD) fat diet for 12 weeks in Wistar rats (*n* = 10/group). XOS increased *F. prausnitzii* growth, having only a minor impact on the GM composition. When supplemented with HFD, XOS ameliorated hepatic steatosis. The underlying mechanisms involved enhanced hepatic β-oxidation and mitochondrial respiration. Nuclear magnetic resonance (^1^H-NMR) analysis of cecal metabolites showed that, compared to the HFD, the LFD group had a healthier cecal short-chain fatty acid profile and on the HFD, XOS reduced cecal isovalerate and tyrosine, metabolites previously linked to NAFLD. Cecal branched-chain fatty acids associated positively and butyrate negatively with hepatic triglycerides. In conclusion, XOS supplementation can ameliorate NAFLD by improving hepatic oxidative metabolism and affecting GM.

## 1. Introduction

In western countries, around 30% of the normal weight and up to 90% of the obese population are estimated to suffer from non-alcoholic fatty liver disease (NAFLD) [1]. NAFLD is defined as excessive fat accumulation in the liver without the patient drinking excessive alcohol or using steatogenic drugs. NAFLD can be categorized into simple hepatic steatosis, which is diagnosed as the presence of fat accumulation without histological or biochemical injuries, and non-alcoholic steatohepatitis (NASH), which is characterized by hepatic steatosis, inflammation and damage of the hepatocytes [2,3]. 

Increasing evidence shows that the pathogenesis of NAFLD is associated with environmental, host genetic and physiological factors [4], such as increased lipid storage [5,6,7] and mitochondrial dysfunction [7]. Frequently, dietary factors and excessive caloric intake are involved in the pathogenesis of NAFLD, and they are also important determinants of the gut microbiota (GM) composition of the host [8]. The GM refers to the trillions of tiny microbial cells inhabiting the gastrointestinal tract that break down the macromolecules and nutrients from the ingested food. Complex interactions between dietary factors and microorganisms are known to dictate the beneficial or detrimental effects on the host health [8]. Prominently, recent studies have highlighted the importance of gut-derived signals [9,10] and the entity of the GM in the pathogenesis of NASH and NAFLD. In the GM of NAFLD patients, for instance, over-represented *Gammaproteobacteria* [11,12] and the genera *Lactobacillus*, *Dorea*, *Robinsoniella* and *Roseburia* [13] have been found compared to healthy controls. Controversially, either low abundance [14] or high abundance [15] of the phylum *Bacteroidetes* has been detected in NASH patients. Another study reported enriched *Fusobacteria*, *Lachnospiraceae, Enterobacteriaceae*, *Erysipelotrichaceae* and *Streptococcaceae* in NAFLD patients [16]. These studies indicate that there is no single microbial taxon always positively or negatively associated with liver fat, which, however, may be influenced by the age, gender and geographic location of the study cohorts that knowingly affect the composition of the GM [17,18,19,20].

The involvement of the GM in NAFLD has led to the evaluation of possible therapies that either use health-beneficial microbes or target the GM of the host. We have shown that intragastric administration of *Faecalibacterium prausnitzii*, a commensal member of the GM with known anti-inflammatory properties [21,22], prevented NAFLD in mice [23]. The rationale for investigating this bacterium in the mice model stemmed from our human study, which showed a reverse association of *F. prausnitzii* abundance with hepatic fat content [24]. Thus, we hypothesized that its low abundance might partly contribute to the diseased phenotype. Our study was in agreement with another study that detected under-represented *F. prausnitzii* in NASH patients [25].

Our previous findings thus tentatively suggested that NAFLD might be partly relieved with *F. prausnitzii.* Unfortunately, not all potentially “therapeutic” bacteria are accepted for human use. Therefore, we searched for alternative, nutrition-based tools to increase the natural abundance of *F. prausnitzii* to treat NAFLD. Such effective tools are known to be probiotics and prebiotics, for instance [26]. A probiotic refers to a live microorganism that positively affects the health of the host, while a prebiotic is defined as a selectively fermented dietary component that cannot be digested as such but serves as food for the GM. A prebiotic thereby causes specific changes in the composition and/or functions of the GM, conferring beneficial effects upon the health of the host. Carbohydrates, such as dietary fiber, are potential prebiotics. These include xylo-oligosaccharides (XOS), fructo-oligosaccharides (FOS), galacto-oligosaccharides (GOS), isomalto-oligosaccharides (IMO), transgalacto-oligosaccharides (TOS) and soybean oligosaccharides (SBOS) [26]. Previously, a 2.8 g daily dose of XOS, isolated from corncobs, was shown to induce an increase in the abundance of *Faecalibacterium* species in a group of seven healthy humans [27]. On the contrary, two in vitro studies failed to show any stimulatory effect of XOS on *F. prausnitzii* growth [28,29]. Moreover, a human study did not show effects of XOS on *F. prausnitzii* but a slightly improved glucose tolerance was achieved in pre-diabetic subjects when they consumed 2 g of XOS daily, for eight weeks [30]. 

In the present study, we first tested in vitro whether *F. prausnitzii* can utilize XOS and how 0.5% XOS affects the growth of *F. prausnitzii*, and we then studied the effects of a prebiotic dose of XOS (0.12%) in vivo. In rats, NAFLD was induced with a high-fat diet (HFD), with or without XOS supplementation. In addition, the effects of XOS (or not) were studied in healthy controls, i.e., in normal, low-fat diet (LFD) fed counterparts that did not generate NAFLD.

## 2. Materials and Methods 

### 2.1. In Vitro Cultivations of Faecalibacterium prausnitzii 

In vitro cultivations of *F. prausnitzii* were done in fastidious anaerobe broth (FAB) supplemented with short-chain fatty acids (SCFA) in an anaerobic workstation (WhitleyA35, Don Whitley Scientific, West Yorkshire, UK). The effects of XOS were studied on two commercially available *F. prausnitzii* strains, American type of culture collections (ATCC)^®^-27766™ (Manassas, VA, USA) and DSM A2-165 (Deutsche Sammlung von Mikroorganismen und Zellkulturen GmbH (DSM), Braunschweig, Germany), with three replicates for both. Ten ml cultures of FAB+SCFA media were supplemented or not with XOS as 0.5% (*w*/*v* %) as recommended by the scientific advisors of the manufacturer. XOS was isolated from corncobs (*Zea mays* subsp. *mays*) by hydrolyzing enzymatically. It was donated by the manufacturer, Shandong Longlive Biotechnology LTD, China (95% pure, CAS #87099-0). The growth of *F. prausnitzii* was determined by measuring changes in the optical density at 620 nm with a MultiskanFC photometer (Thermo Fisher Scientific, Waltham, MA, USA) after 19, 20.5, 22 and 23.33 h of cultivations.

### 2.2. Animals

Approval for the animal experiment was received from the National Animal Experiment Board of Southern Finland (ESAVI/8805/4.10.07/2017), and the study was performed in accordance with the Guidelines of the European Community Council directives 2010/63/EU and the European Convention for Protection of Vertebrate Animals used for Experimental and other Scientific Purposes (Council of Europe No123, Strasbourg 1985). In the experiment, the Animal Research: Reporting of In Vivo Experiments (ARRIVE) guidelines were followed. Male Wistar rats, 10–12 weeks of age, were purchased from Charles River, Europe. Wistars were chosen because our pilot studies indicated that they harbor *F. prausnitzii* in their GM while Sprague Dawley, for instance, do not (data not shown). Upon arrival, the rats were allowed to habituate to the environment for two weeks. During the whole experiment, the rats were single-housed. The rats were divided randomly into four dietary treatment groups (*n* = 10/group): (1) high-fat diet (HFD, with 60% of energy from fat), (2) HFD supplemented with XOS (0.12%, HFD+XOS), (3) control = low-fat diet (LFD, with 10% of energy from fat), and (4) LFD supplemented with XOS (0.12%, LFD+XOS). It is of note that the control feed contained a standard amount of fat available in the rodent feed, but for clarity and to distinguish it from the high-fat group, it is termed LFD. In humans, a daily dose of 2.8 g, i.e., 0.035 g/kg, of XOS yielded induction of *F. prausnitzii* abundance [27] and therefore we targeted a similar dose level in vivo. The average dose of XOS for the rats in our study was 0.05 g/kg. XOS was isolated as described above and donated by Shandong Longlive Biotechnology (95% pure, CAS #87099-0). All irradiated diets were purchased as custom-made pellets from Labdiet/Testdiet, UK. The animals had food and water ad libitum and were maintained in a 12/12 h light/dark cycle in an enriched environment at animal facilities of the University of Jyväskylä. At the beginning of the 12-week diet intervention, all rats were ~15 weeks of age.

### 2.3. Indirect Metabolic Measurements 

The indirect measures of metabolism were analyzed from respiratory gases with oxygen and CO_2_ as well as with an analyzer for the capacitive water vapor partial pressure (Promethion^®^GA3, Sable Systems, Las Vegas, NV, USA). The air flow was controlled and measured by two multi-channel mass flow generators (FR8, Sable Systems). The incurrent flow rate was set at 3500 mL/min. The data acquisition was synchronized by MetaScreen^®^ and the raw data were processed with ExpeData^®^ software (Sable Systems). The ExpeData carries out all data transformation, calculating the respiratory quotient (RQ) as the ratio of CO_2_ production over O_2_ consumption and energy expenditure by utilizing the equation of Weir: Kcal/h = 60 × (0.003941 × VO_2_ + 0.001106 × VCO_2_) [31].

### 2.4. Measurement of Weight, Body Composition and Food Intake

During the study, the animals were weighed once a week always at the same time of day on an electronic scale. The food intake was measured once a week by weighing the consumed feed over 24 h. The body composition was determined with dual energy x-ray absorptiometry (DXA, Prodigy; GE Lunar Corp., Madison, WI, USA) under isoflurane anesthesia before and after the diet intervention.

### 2.5. Necropsy, Blood Analyses, Tissue Collection and Histology

After the 12-week diet intervention, the non-fasted rats were anesthetized with a mixture of air and CO_2_ and euthanized by drawing the blood by cardiac puncture. KONELAB 20XTi analyzer (Diagnostic Products Corporation, Los Angeles, CA, USA) was used to determine the serum levels of glucose, triglycerides, free fatty acids, glycerol, cholesterol, LDL (low-density lipoprotein) cholesterol, HDL (high-density lipoprotein) cholesterol, aspartate aminotransferase (AST) and alanine aminotransferase (ALT). Serum insulin was measured with IMMULITE analyzer (Siemens healthineers, Erlangen, Germany). Serum cytokines were analyzed with 9-plex cytokine ELISA kit (#110449RT), Quansys and Q-View software (Quansys Biosciences, Logan, UT, USA), as specified in the instructions of the manufacturer. The detection limits for the cytokines were as follows: interleukin (IL)-1a, 8.55 pg/mL; IL-1β, 3.58 pg/mL; IL-2, 2.74 pg/mL; IL-4, 0.45 pg/mL; IL-6, 1.4 pg/mL; IL-10, 0.26 pg/mL; IL-12, 0.41 pg/mL; interferon (IFN)-γ, 33.71 pg/mL and tumor necrosis factor (TNF)-α, 2.72 pg/mL. 

The medial lobe of the liver was harvested and, after excising the samples for the mitochondrial respiration analysis and histology, the rest of the medial lobe was immersed in liquid nitrogen and stored at −80 °C. For the subsequent analyses of enzyme activity, fat content and mRNA expression, the frozen medial lobe was pulverized in liquid nitrogen. For histology, a piece of the liver was snap-frozen in cooled isopentane (−150 °C) and stored at −80 °C. Neutral lipids were visualized from 10 µm cryosections with Oil Red O staining. Paraformalin-fixed sections were rinsed with H_2_O, stained for 15 min with freshly prepared Oil Red O solution (Merck, Kenilworth, NJ, USA) and rinsed with 60% isopropanol to avoid over-staining. The sections were counterstained with Mayer’s hematoxylin and scanned with NanoZoomer microscope (Hamamatsu, Japan). The amount of hepatic fat was scored by two blinded experimenters. 

To estimate the amount of liver fibrosis, 10 µm paraformaldehyde-fixed cryosections were stained with Sirius Red. The sections were first stained with Weigert’s hematoxylin (Sigma-Aldrich, St Louis, MO, USA), then washed with tap water and stained in Sirius Red (Sigma-Aldrich)–picric acid (Sigma-Aldrich) solution. Finally, the sections were immersed twice in acidified water and cover-slipped with Depex following dehydration and clearing in ethanol and xylene. Then, the sections were scanned with NanoZoomer and the amount of fibrosis was scored from the scanned images by two blinded experimenters. The scoring was based on Ishak grade [32]. Briefly, 0 is no fibrosis; 1 is fibrous expansion of some portal areas and short fibrous septa; 2 is fibrous expansion of most portal areas and short fibrous septa; 3 is fibrous expansion of most portal areas and occasional portal to portal bridging; 4 is fibrous expansion of portal areas and marked portal to portal as well as portal to central bridging; 5 is marked bridging and occasional nodules, and 6 is probable or definite cirrhosis. 

For the histology of the gut, ~10 mm of proximal colon was cut right after the cecum, the colon contents were collected (see below) and the rest of the feces was washed out with phosphate-buffered saline (PBS). The collected tissue was fixed with 4% paraformaldehyde for 48 h, washed twice with PBS and then stored in 70% ethanol at 4 °C. The histopathological scoring of the proximal colon was done by a veterinarian and expert toxicopathologist (author J.L.). For the assessment, the formalin-fixed proximal colon samples of approximately 1 cm of length were trimmed into 4–6 pieces and embedded in paraffin transversally. The paraffin blocks were cut at 3 µm, resulting in 4–6 transversal sections per colon sample. The sections were stained with hematoxylin and eosin. The microscopic findings were classified with standard pathological nomenclature and severities of inflammatory activity, mucosal atrophy and crypt hyperplasia were graded on a suitable scale (generally 1 to 5 as minimal, mild, moderate, marked or severe). The grades of severity for the microscopic findings were subjective; minimal was the least extent discernible and severe was the greatest extent possible. The grading was based on both the severity and extent (focal, extensive and diffuse) of the affection. The crypt hyperplasia was not assessable in some cases. A detailed description of the scoring is provided in the Appendix A.

To visualize intestinal tight junctions, the 3 µm sections were deparaffinized and boiled in 0.01 M sodium citrate (pH 6.0) for antigen retrieval and blocked with 10% goat serum. The intestinal tight junctions were stained with anti-tight junction protein-1 antibody (Tjp1, Novus Biologicals, Littleton, CO, USA) and visualized by labeling with anti-rabbit 647 Alexa Fluor (Invitrogen, Carlsbad, CA, USA). The nuclei were made visible with DNA-stain 4′,6-diamidino-2-phenylindole (DAPI, 1:2000), and then the sections were cover-slipped with Mowiol-mounting media. The labeled sections were imaged with a confocal microscope (Carl Zeiss LSM 700) and the signal intensities were counted with Image J. From each rat, four randomly selected areas were imaged, with four tiles per area (1184 µm × 1184 µm). The fluorescence intensity was normalized to the intensity of DAPI. 

### 2.6. Collection of Gut Contents, DNA Extraction and Real-Time Quantitative PCR

The contents of the proximal colon and cecum were collected at necropsy, snap-frozen in liquid nitrogen and stored at −80 °C. The total DNA was extracted from ~100 mg of the colon and cecum contents with Stool Extraction Kit and semi-automated GenoXtract (Hain Lifescience GmbH, Nehren, Germany), accompanied by bead-beating in 1.4 mm ceramic bead tubes to ensure better homogenization.

Real-time quantitative PCR (qPCR) was done using DNA extracted from the cecum and *F. prausnitzii* 16S rRNA-targeted primers, as has been described previously [33]. First, in the traditional PCR, pure cultures of *F. prausnitzii* and *F. prausnitzii* 16S rRNA-targeted primers were used. The PCR program was as follows: pre-incubation at 95 °C for 10 min, 40 cycles at 95 °C for 30 s, 60 °C for 1 min and 72 °C for 30 s, followed by a final extension at 72 °C for 8 min (Veriti 96 Well Thermal Cycler, Applied Biosystems, Foster City, CA, USA). The expected size of the PCR product was 140 base pairs. After obtaining and purifying the PCR products, a dilution series for the qPCR standard curve was made by pooling the positive PCR products obtained in the traditional PCR. The concentration of the pooled DNA fragments was measured with NanoDropND-1000 spectrophotometer (ND, Nanodrop Technologies Inc., Wilmington, DE, USA). Taking into account that the molecular weight of one DNA base pair is 660 g/mol and that the size of the PCR product was 140 base pairs, the concentration of DNA fragments in the pool could be calculated based on the DNA concentration: DNA concentration (ng/µL)/molecular weight of one PCR fragment (g/mol) = fragment concentration. Then, by multiplying the fragment concentration (mol/μL) with Avogadro’s number (6.0221415 × 1023), the number of fragments per one μL was obtained. Then, a dilution series from 10^10^ to 10^1^ was done and used as a standard curve in the subsequent qPCR. The qPCR results are shown as *F. prausnitzii* gene copy number/gram of cecal content used to extract the DNA. However, it should be noted that the gene copy number is not likely to be representative of the absolute bacterial cell numbers of *F. prausnitzii* in the samples, because it is not known how many gene copies of 16S rRNA each single *F. prausnitzii* cell has.

### 2.7. 16S rRNA Gene Sequencing and Processing of the Sequence Data

For the microbial community analysis, rRNA gene was amplified using primers 515F-Y (GTGYCAGCMGCCGCGGTAA) and 806R (GGACTACHVGGGTWTCTAAT) that target the V4 region of the subunit ribosomal *RNA* (SSU rRNA) gene. In the first PCR, the reaction consisted of 1xMaxima™ SYBR Green qPCR Master Mix (Thermo Fisher Scientific, Waltham, MA, USA), 0.5 µM of primer and 20 ng of DNA template. The PCR program was as follows: 10 min initial denaturation at 95 °C, 30 cycles at 94 °C 30 s, +52 °C 60 s and 72 °C 60 s, followed by a final extension at 72 °C for 5 min (C1000 ThermalCycler, Bio-Rad Laboratories, Hercules, CA, USA). After this, Ion Torrent PGM sequencing adapters and barcodes were added to the ends of the PCR product by using one µL of the PCR product as a template in the second PCR. In this second PCR, 10 cycles were performed using linker and fusion primers (0.05 µM of M13_515F-Y, 0.5 µM of IonA_IonXpressBarcode_M13 and P1_806R), the other conditions being identical to the first PCR. The sequencing was done using Ion Torrent PGM (Thermo Fisher Scientific). The PCR products were purified with AMPure XP (Beckman Coulter, Brea, CA, USA), quantified with PicoGreen (Quant-iT™ PicoGreen™ dsDNA Assay Kit, Thermo Fisher Scientific, Waltham, MA, USA) and pooled in equimolar quantities for sequencing on Ion Torrent PGM using Hi-Q View OT2 Kit for emulsion PCR, Hi-Q View Sequencing Kit for the sequencing reaction and Ion 318 Chip v2 (Thermo Fisher Scientific). 

Quality filtering and clustering to the operational taxonomic units (OTUs) at the 97% similarity level of the 16S rRNA gene sequences was carried out using the CLC Microbial Genomics Package (Qiagen, Hilden, Germany). After preprocessing, the dataset contained 5.0 million reads—on average, 64,544 ± 15,932 reads per sample and 12,700 unique OTUs. The rRNA gene sequences were classified using the SILVA SSU Ref database (v132, 99%). First, the GM diversity was analyzed from the cecal and colon contents, separately. Because no differences in the distribution of evenness between species, i.e., alpha-diversity of the GM, were observed between the two intestinal compartments (see Results, Section 3.2), the sequence data of the cecum and colon samples were pooled for further analyses of the GM composition.

### 2.8. Extraction, Identification and Analyses of Cecal Metabolites 

Ice-cold PBS (pH 7.4) was mixed with the cecal content at a ratio of 1:2 and vortexed for 5 min to extract cecal metabolites. The extract was then centrifuged at 15,000× *g* for 15 min at 4 °C. A part of the resultant supernatant was mixed with 10% Chenomx standard solution (5 mM deuterated DSS (DSS-d6) and sodium azide in D_2_O) and vortexed for 15 s. Then, 180 µL of the prepared mix was placed into 3 mm nuclear magnetic resonance (NMR) tubes. The spectra were recorded using a 600 MHz Bruker AVANCE-III NMR spectrometer, which was equipped with TCI Prodigy CryoProbe (Bruker BioSpin AG, Fällanden, Switzerland). 

The recording of the NMR spectra was performed using a 600 MHz Bruker AVANCE-III NMR spectrometer (Bruker BioSpin AG, Fällanden, Switzerland) that was equipped with a TCI Prodigy CryoProbe. A 128-scan Carr–Purcell–Meiboom–Gill sequence (CPMG) pulse was applied to acquire the spectra consisting of 128 k data points at a spectral width of 10 kHz, at 25 °C, with an acquisition time of 6.82 s. To acquire the *J*-resolved (JRES) spectra, the parameters that we used were 16 scans, 1 k data points, 128 increments, 2 s relaxation delay and a spectral width of 16 ppm in dimensions. The heteronuclear single quantum coherence (HSQC) spectra were acquired using 32 scans, 2 k data points, 128 increments, 2 s relaxation delay and spectral width of 16 ppm and 165 ppm in the proton and carbon dimensions, respectively.

### 2.9. Processing of the Cecal Metabolite Data and Multivariate Data Analysis 

The metabolites were identified and quantified with the aid of Chenomx NMR Suite 8.3 Professional (Chenomx Inc., Edmonton, Alberta, Canada), Human Metabolome Database (HMDB, http://www.hmdb.ca), and Biological Magnetic Resonance data Bank (BMRB, http://www.bmrb.wisc.edu). Two-dimensional JRES and HSQC spectra were used in confirming the identities of the metabolites.

The ^1^H NMR spectra were manually phased, baseline and shim-corrected and referenced to trimethylsilylpropanoic acid (TSP) at 0.00 ppm using Chenomx NMR Suite. The misaligned regions of the spectral set were corrected using *icoshift*. After removal of the residual water (4.68–4.88) region, the spectra were subjected to total area normalization and then reduced to variable sized bins of width ranging from 0.018 to 0.04 ppm. This resulted in a dataset comprising 40 observations and 157 variables. The pareto-scaled dataset was then subjected to unsupervised (PCA) and supervised multivariate analyses using (PLS-DA and OPLS-DA) using SIMCA-P 14.1 (Umetrics, Sartorius Stedim Biotech, Umeå, Sweden). The validation of the supervised multivariate models was achieved by 100 permutation test, estimation of explained variation (R2Y (cum)), predictive ability (Q2Y (cum)) and CV-ANOVA values.

### 2.10. Measurement of the Liver Fat Content and 3-Hydroxyacyl-CoA Dehydrogenase Activity 

To extract the total lipids, a pulverized sample of the medial lobe of the liver was analyzed with KONELAB 20XTi, as described previously by us [34]. To analyze the activity of 3-hydroxyacyl-Coenzyme A (CoA) dehydrogenase 8 (β-HAD), ~20 mg of pulverized liver was homogenized with cold lysis buffer (10 mM Tris-HCl, 150 mM NaCl, 2 mM ethylenediaminetetraacetic acid (EDTA), 1% Triton X-100, 10% glycerol and 1 mM dithiotreitol (DTT)) that contained protease and phosphatase inhibitors (Sigma Aldrich, St Louis, MO, USA) using TissueLyzer (Qiagen, Valencia, CA, USA). After separating the insoluble material by centrifugation at 12,000× *g*, β-HAD activity was determined from the supernatant with KONELAB 20XTi. For the measurement, the supernatant was mixed with a solution that contained 50 mM Triethanolamine-HCl (pH 7.0), 4 mM EDTA, 0.04 mM NADH and 0.015 mM S-Acetoacyl CoA.

### 2.11. Measurement of Hepatic Mitochondrial Functions with High-Resolution Respirometry to Analyze the Rate of Hepatic Glucose Metabolism

A freshly collected sample of the medial lobe of the liver (15–20 mg) was homogenized in 0.5 mL Mir05-medium with PBI-Schredder high-resolution respirometry set (Oroboros instruments, Innsbruck, Austria). Shredding was done for 10 s 1-level + 5 s 2-level. The wash of the shredding pipe was performed by applying three times 0.5 mL of Mir05-medium. Medium was collected into a clean tube with the homogenate, and the final volume was set to 5 mL with Mir05-buffer. An aliquot of 0.8 mL of the homogenate and 1.5 mL of Mir05-medium was transferred to the Oroboros O2k-Respirometer (Oroboros instruments). DatLab software (Oroboros instruments) was used to record oxygen concentration (µM) and oxygen flux per tissue wet mass (pmol O2•s^−1^•mg^−1^). Oroboros program was performed as follows: (1) Pyruvate, malate and glutamate (5 mM, 2 mM, 10 mM) were used as initial substrates but without adenosine diphosphate (ADP) leak respiration being measured. (2) The addition of ADP and Mg^2+^ (4 mM, 2.4 mM) started oxidative phosphorylation through mitochondrial complex I (CI). (3) Cytochrome c (10 µM) was added in order to monitor unwanted mitochondrial degradation. (4) Succinate (10 mM) was added as a complex II substrate (CI+II). (5) The maximal capacity of the electron transport system was measured by adding carbonyl cyanide m-chlorophenyl hydrazone (CCCP) (0.5–2.5 µM; until max O_2_ consumption was reached). (6) The complex I was inhibited with rotenone (0.5 µM) and (7), complex III with antimycin A (2.5 µM) and finally only the residual oxygen consumption (ROX) was left. ROX was subtracted from all other oxygen flux values and all values were expressed as normalized to the wet tissue mass.

### 2.12. qPCR of Liver and Adipose Tissues

The total RNA from the pulverized liver as well as mesenteric and epididymal adipose tissues was extracted using Tri reagent (Thermo Fisher Scientific) according to the protocol provided by the supplier. The RNA was reverse transcribed into cDNA using High Capacity cDNA synthesis kit as specified in the instructions of the manufacturer (Applied Biosystems). The real-time PCR analysis was done according to the Minimum Information for Publication of Quantitative Real-Time PCR Experiments (MIQE) guidelines using in-house designed primers (from Invitrogen), iQ SYBR Supermix and CFX96™ Real-time PCR Detection System (Bio-Rad Laboratories, Hercules, CA, USA). The primer sequences were as follows: stearoyl conenzyme saturase 1 (SCD1) forward 5′CCTCATCATTGCCAACACCAT3′ and reverse 5′AGCCAACCCACGTGAGAGAA3′; diacylglycerol O-acyltransferase 2 (DGAT2) forward 5′GGGTCCAGAAGAAGTTCCAGAAG3′ and reverse 5′CCAGGTGTCAGAGGAGAAGAG3′; IL1beta forward 5′CACAAAAATGCCTCGTGC3′ and reverse 5′TGCTGATGTACCAGTTGGG3′; leptin forward 5′AGCAGTGCCTATCCAGAAGT3′ and reverse 5′TTCTCCAGGTCATTGGCTAT3′; adiponectin forward 5′AATCCTGCCAGTCATGAAG3′ and reverse 5′CATCTCCTGGGTCACCCTTA3′.

### 2.13. Statistical Analyses

The statistical analyses, except for the GM and their metabolites, were performed with IBM SPSS Statistics v24 for Windows (SPSS, Chicago, IL, USA). The main effects of the diet and XOS were determined using a general linear model or mixed model analysis. The group differences were analyzed with ANOVA. Kruskal–Wallis or median test was used if the data were not normally distributed according to the Shapiro–Wilk test in SPSS. For the repeated measures, we used linear mixed model. Type III tests of fixed effects with Sidak’s adjustment for multiple comparisons were used. Cohen’s *d* was used to estimate differences between the groups in *F. prausnitzii* abundance because of the high inter-individual variation. The associations between the variables were studied with Spearman’s correlation coefficient. 

The alpha-diversity of the GM was quantified with Shannon index using the *microbiome* R/Bioconductor package. The analysis of beta-diversity was based on Bray–Curtis distance and PERMANOVA [35] for significance testing (*adonis* function in the *vegan* R package). The taxonomic groups were compared based on DESeq2 [36] using the *phyloseq* R/Bioconductor package, including Benjamini–Hochberg correction for multiple testing. The statistical significance was set at *p* < 0.05 after the multiple testing corrections.

The univariate analysis of the concentrations of cecal metabolites was performed using Graph Pad Prism 8.0 (GraphPad Software, San Diego, CA, USA). The normal distribution of the data was tested with the Shapiro–Wilk test. The differences between the groups were assessed with the parametric one-way ANOVA for normally distributed variables and with Tukey’s multiple comparison test or non-parametric Kruskal–Wallis test for the non-normally distributed variables with Dunn’s multiple comparisons. The statistical significance was set at *p* < 0.05.

## 3. Results

### 3.1. XOS Increased the Growth of Faecalibacterium Prausnitzii and Concomitantly Decreased Hepatic Fat Content Due to Enhanced Fat and Glucose Metabolism

We first tested in vitro whether *F. prausnitzii* can utilize XOS and how XOS affects its growth. Compared to the control treatment, XOS increased the growth of *F. prausnitzii* in vitro (F (1, 16.9) = 64.9, *p* < 0.001). The effects of time (F (3, 25.6) = 5.4, *p* = 0.005) and cell line (F (1, 16.9) = 59.6, *p* < 0.001) were also significant. Thus, the difference between the control and XOS treatment was significant at each time point (after 19, 22, 23.3 h of stimulation) in the DSM A5-165 strain (*p* < 0.05) but not in the ATCC-27766 strain (Figure 1a). In rats, after 12 weeks of dietary XOS supplementation, *F. prausnitzii* abundance was increased compared to the HFD (Cohen’s d = 0.2) and LFD (Cohen’s d = 0.2) without XOS (Figure 1b). Real-time quantitative PCR was used to analyze *F. prausnitzii* abundance in rats, because unfortunately, the primers used in the 16S rRNA gene sequencing did not catch *F. prausnitzii*.

As expected, the biochemical measurement of the hepatic triglyceride content showed that the HFD increased triglycerides (F (1, 6.0) = 46.4, *p* < 0.001). XOS decreased triglycerides on the HFD (F (1, 0.8) = 6.5, *p* = 0.017), while when supplemented with LFD, XOS increased hepatic triglycerides (Figure 2a). A similar interactive effect of XOS was found on the total hepatic cholesterol (F (1, 0.2) = 15.2, *p* = 0.001), although in group comparisons only the LFD+XOS had significantly higher hepatic cholesterol content than the LFD (*p* = 0.015, Figure 2a). To confirm the findings on the content of hepatic triglycerides, the medial lobe of the liver was analyzed also histologically. Oil Red O staining of the frozen liver sections showed that XOS supplementation decreased hepatic neutral lipids compared to the HFD in rats (Figure 2b). Based on the Sirius Red staining, there was no pronounced progression to hepatic fibrosis in any of the diet groups or differences in the fibrosis score between the groups. Yet, there was, to some extent, fibrous expansion in portal areas and short fibrous septa in all groups, which could be due to having feed ad libitum. Examples of typical stainings are shown in Appendix A.

To assess whether the decreased hepatic fat content was due to increased fat oxidation, we determined the activity of β-HAD, the rate-limiting enzyme of fatty acid β-oxidation. XOS increased the hepatic activity of β-HAD on the HFD (F (1, 36) = 4.5, *p* = 0.041) (Figure 3a).

We then analyzed hepatic glucose metabolism using high-resolution respirometry. In the hepatic mitochondria, the HFD decreased maximal electron transport (*p* = 0.034) (Figure 3b). The HFD also lowered the maximal electron transport capacity theoretically available for the oxidative phosphorylation (*p* = 0.023), and the reserve electron transport capacity beyond the oxidative phosphorylation through complex I (*p* = 0.019) as well as complexes I and II (*p* = 0.013) (Figure 3c). XOS supplementation seemed to ameliorate these effects of the HFD on mitochondrial respiration (Figure 3c). Compared to the HFD, HFD+XOS had increased respiratory capacity available for the production of ATP through the electron flow from complex I (*p* = 0.023, Figure 3c) and improved coupling of electron transport through complex I and oxidative phosphorylation (*p* = 0.041, Figure 3d).

The HFD decreased but XOS had no effects on the hepatic mRNA expression of SCD1 (effect of diet: F (1,36) = 97.5, *p* < 0.001) and DGAT2 (effect of diet: F (1,35) = 8.7, *p* = 0.006), which are involved in lipogenesis and the synthesis of triglycerides, respectively (Figure 4).

### 3.2. The Diets Did Not Affect the Diversity of the Gut Microbiota, but Minor Differences Were Found in the Relative Abundances of Three Microbial Genera

Despite the over 12,000 OTUs, the proximal colon and cecum of the rats were dominated by a few phyla and genera. Bacteroidetes (51.6% of all sequences) and Firmicutes (39.7%) were the dominating bacterial phyla, followed by Verrucomicrobia (3.6%) and Proteobacteria (2.3%). The families Tannerellaceae (19.1%, only genus Parabacteroides), Rikennellaceae (14.6%, mostly genus Alistipes) and Muribaculaceae (10.0%, several genera) explained the dominance of Bacteroidetes, and the families Ruminococcaceae (16.4%) and Lachnospiraceae (13.3%) were the dominating sequences belonging to Firmicutes. The hierarchical clustering analysis suggested two main clusters among the rat groups based on their GM profiles. The dietary fat explained the clusters regardless of XOS supplementation (data not shown). However, the diets did not affect the alpha-diversity (Figure 5a) or the beta-diversity (Figure 5b) of the colon or cecum GM. The GM composition did not differ between the groups at phylum (Figure 6a) or family level, including the Ruminocaccaceae family, to which *F. prausnitzii* belongs (data not shown). At genus level, *Dubosiella* and uncultured members of Christensenellaceae were lowest and the Prevotellacaeae NK3B31 group highest in the HFD+XOS (*p* = 0.01 for all, Figure 6b). The GM of the HFD tended to have higher relative abundance of Prevotellacaeae UCG-10 (*p* = 0.06), and LFD groups had higher abundance of Anaerostipes (*p* = 0.10) (data not shown).

### 3.3. XOS Improved HFD-Induced Intestinal Damage and Inflammatory Markers in the Proximal Colon, but Did Not Affect Intestinal Tjp1, Hepatic IL1β mRNA or Systemic Inflammation

In the histopathological examination of the proximal colon, an interactive effect of the diet and XOS was found on the surface epithelial injury (F (1, 35) = 35.0, *p* < 0.001), Goblet cell hyperplasia (F (1, 35) = 12.6, *p* = 0.001), crypt length distortion (F (1, 35) = 31.0, *p* < 0.001) and injury score (F (1, 35) = 90.4, *p* < 0.001) (Figure 7). XOS diminished the epithelial injury caused by the HFD, while on the LFD, XOS increased the injury (F (1, 35) = 18.9, *p* < 0.001) (Figure 7). On the LFD, XOS also increased monocyte aggregates, i.e., leukocyte infiltration to the lamina propria (F (1, 36) = 5.8, *p* = 0.021) but had no effects on lymphocyte aggregates in lamina propria, i.e., cryptopatches or large granular lymphocytes (Figure 7).

However, despite these important changes in the epithelia and the fact that HFD is known to compromise gut integrity and, conversely, prebiotic nutrients are known to enhance it, the histological analysis of the tight junctions showed no differences between the groups (Figure 8).

As mentioned, F. prausnitzii is reported to exert anti-inflammatory functions, for which we were interested in studying whether its increased abundance was associated with decreased systemic inflammation. Of the nine cytokines that were analyzed, only three were detected in rats, namely interleukin(IL)-10, IL-12 and tumor necrosis factor-α (TNFα). XOS supplementation and thus higher abundance of F. prausnitzii were not associated with changes in the levels of these cytokines (Figure 9). TNFα was under the detection limit, i.e., <2.72 pg/mL in the HFD groups, and was detected in three out of ten rats in the LFD groups. Thus, the effect of diet on TNFα was not definable. The HFD diminished the serum level of anti-inflammatory IL-10 (F (1, 33) = 18.0, *p* < 0.001) (Figure 9). Due to the importance of pathways that activate IL1 family cytokines in the development of non-alcoholic fatty liver disease (NAFLD), we also analyzed the expression of IL1β mRNA in the liver. No effects of the diets were found on the mRNA expression (Figure 9).

### 3.4. The HFD Had Harmful Effects on the Cecal SCFA Profile, and on the HFD, XOS Decreased the Levels of Cecal Isovalerate and Tyrosine 

The metabolomic analyses of the cecum contents detected a clear separation between the HFD and LFD based on the component t [1] in OPLS-DA (Appendix A). The model represented a high goodness of fit (*R*^2^X_(cum)_ = 0.552 and *R*^2^Y_(cum)_ = 0.9) and optimum predictive ability (Q^2^ = 0.841). The metabolites responsible for the separation of the HFD and LFD were identified by combining the information from the S-loading plot (Appendix A), column loading plot as well as Variable Importance in Projection (VIP) data. The details and comparisons are compiled in the Appendix A. The unsupervised PCA did not differentiate between the metabolites of the HFD and HFD+XOS (Appendix A) or between the LFD and LFD+XOS (Appendix A). As expected, the LFD was characterized by higher cecal levels of SCFAs (Table 1). The dietary fat mostly explained the clustering of the cecal metabolites (Figure 10a). In general, XOS lowered the level of cecal tyrosine (F (1, 36) = 7.7, *p* = 0.009) (Figure 10b). XOS and diet had an interactive effect on isovalerate (F (1, 36) = 6.0, *p* = 0.012), and on the HFD, XOS had a significant decreasing effect on it (F (1, 36) = 8.2, *p* = 0.007) (Figure 10b). Several cecal metabolites, including the SCFAs, associated negatively with the content of hepatic triglycerides (Figure 10c). Metabolites that are known to boost oxidative metabolism, such as nicotinate, butyrate and 2-oxoglutarate, associated positively with hepatic oxidative phosphorylation and negatively with triglyceride content (Figure 10c). The group-wise comparisons of the metabolites are shown in Appendix A.

### 3.5. The HFD Decreased Energy Expenditure in Rats while the Diet and XOS Had an Interactive Effect on the Energy Expenditure

The HFD lowered the average energy expenditure independently of the time of day (main effect of diet: daytime, F (1, 36) = 19.5, *p* < 0.001; night, F (1, 35) = 20.9, *p* < 0.001; Figure 11a). XOS and diet had an interactive effect on the nighttime lowest resting energy expenditure (F (1, 35) = 7.9, *p* = 0.008) and the tendency was similar during daytime (F(1, 36) = 3.5, *p* = 0.071, Figure 11a). That is, at night, HFD lowered resting energy expenditure but XOS enhanced it, whereas on the LFD, XOS diminished the resting energy expenditure. Compared with the HFD, LFD had higher measured lowest diurnal energy expenditure over 30 min (*p* = 0.001, Figure 11a). At nighttime, the LFD had higher average energy expenditure than the HFD (*p* = 0.01, Figure 11a). Consequently, the mean produced CO_2_ was higher in the LFD than the HFD during daytime (*p* < 0.001) and at night (*p* = 0.001, Figure 11b). The HFD diminished the production of CO_2_ at nighttime (F (1, 36) = 198.5, *p* < 0.001) and during daytime (F (1, 34) = 164.7, *p* < 0.001 Figure 11b). Further, the diet had a significant effect on all measured diurnal and nocturnal RQs, the RQ values being higher in the LFD than in the HFD groups. The group differences are shown in Figure 11c. In addition, the interactive effect of XOS and diet was significant on RQs at daytime and nighttime (*p* < 0.001 for all, Figure 11c).

### 3.6. XOS Did Not Affect Typical NAFLD-Associated Markers or Body Composition but Had Interactive Effects on the Energy Intake

XOS did not affect serum levels of triglycerides, free fatty acids, total cholesterol, LDL, HDL, glycerol, AST or ALT (Table 2). The diet had an effect on the serum glycerol, AST and ALT (*p* < 0.001 in each), the HFD increasing their levels. The HFD also tended to increase free fatty acids (F (1, 36) = 3.8, *p* = 0.059), while XOS tended to increase triglycerides (F (1, 35) = 3.2, *p* = 0.083). The LFD groups weighed less than the HFD groups during the diet intervention (Appendix A). XOS did not influence body fat %, total fat mass, total tissue mass or the amount of epididymal or mesenteric fat (Appendix A). XOS enhanced the daily energy intake on the HFD and diminished the energy intake on the LFD (*p* = 0.002) (Appendix A).

We also analyzed the expression of adiponectin and leptin mRNA in the mesenteric and epididymal adipose tissues because they have been previously linked to NAFLD. The diet or XOS had no main effects on the expression of adiponectin (AdipoQ) mRNA in the mesenteric adipose tissue, but on the LFD, XOS increased its expression (F (1, 34) = 4.6, *p* = 0.04) (Figure 12). Leptin could not be faithfully detected from the mesenteric adipose tissue. In the epididymal adipose tissue, the diet but not XOS had an effect the expression of leptin (F (1, 36) = 25.5, *p* < 0.001) and AdipoQ (F (1, 36) = 8.9, *p* = 0.005) mRNA, the HFD diminishing the expression of both (Figure 12).

## 4. Discussion

In this study, we show that high-fat-diet-induced NAFLD in rats could be partly treated with dietary, prebiotic xylo-oligosaccharides (XOS). We show that the prebiotic diet had only minor effects on the GM in general. However, XOS slightly increased the abundance of *Faecalibacterium prausnitzii,* whose low relative abundance we have previously found to be associated with high hepatic fat content in humans [24]. Further, *F. prausnitzii* is being considered as one potential next-generation probiotic bacterium [37]. Here, we challenged the rats with high-fat diet or not, and with XOS supplementation or not. The hepatic fat and triglyceride contents were diminished in the XOS-supplemented high-fat diet group, by the XOS enhancing hepatic β-oxidation and mitochondrial respiration. However, no severe hepatic fibrosis was detected in any of the diet groups; thus, the steatosis was not in a very advanced state. Our own previous study in a mice model suggested that NAFLD could be alleviated by administering *F. prausnitzii* intragastrically [23]. Because not all potentially “therapeutic” bacteria are accepted for human use, it is important to find dietary approaches that yield effects similar to the bacteria, as it was done in the present study.

Previously, XOS has been shown to have prebiotic properties by increasing the abundance of beneficial bacteria, such as *Lactobacilli* and *Bifidobacteria* as well as *Faecalibacterium* species [38]. However, *F. prausnitzii* was not explicitly studied. In contrast to our present findings, by studying bacteria in a human colonic simulator, Christophersen et al. found that XOS decreased the growth of *F. prausnitzii* [28]. However, there are methodological differences between the latter and our study that may explain the controversies. Christophersen et al. first stimulated the GM with 3% soy protein to study its detrimental effects on the gut health. Then, the treatment was followed by soy protein with 1% XOS. 

In the present study, XOS improved the HFD-induced colonic injury score and crypt length distortion. Our findings are in accordance with a previous study showing that prebiotic oligosaccharides can prevent HFD-induced intestinal damages [39]. However, XOS did not affect intestinal Tjp1 according to our immunohistological analysis. The latter finding was somewhat surprising because we earlier showed that *F. prausnitzii* administration increased the expression of intestinal *Tjp1* mRNA [23], and others have shown that prebiotic oligosaccharides in general can prevent HFD-induced impairment of gut permeability [39]. Thus, further studies are needed to determine the effects of XOS on other tight junction proteins. 

Independent of the dietary fat, XOS increased the abundance of *Prevotellacaeae* NK3B31 group in rats. Similarly, others have shown that by increasing the amount of dietary fiber by pectin supplementation, the abundance of this bacterial group increased in rats [40]. To our knowledge, *Prevotellacaeae* NK3B31 has not been linked before our study to NAFLD. However, it has been shown to associate negatively with serum triglycerides and LDL cholesterol in type 2 diabetic rats [40]. *Dubosiella* and uncultured members of *Christensenellaceae* were as high in the LFD groups as in the HFD group without XOS supplementation. These were rather unexpected findings, because an obese human cohort has been characterized by a decreased relative abundance of *Christensenellaceae* [41], and a member of this family, *C**hristensenella minuta*, has been suggested to promote a lean host phenotype [42]. Interestingly, in our study, XOS supplementation decreased the abundance of *Christensenellaceae* in the HFD group that had higher body weight.

Despite the increase in the abundance of *F. prausnitzii*, which is considered an important anti-inflammatory bacterium [21,22], no differences in the levels of serum cytokines IL-10, IL-12 or TNFα were found in response to the XOS supplementation. However, it should be noted that we determined the levels of the cytokines only at the end of the diet intervention; thus, we may have missed whether their concentrations were elevated at some other time point during the 12-week diet intervention. Nevertheless, the HFD lowered serum levels of anti-inflammatory IL-10. This result is in agreement with the lower levels of IL-10 in NAFLD patients than in healthy controls [43]. Regardless of the systemic cytokines, XOS seemed to reduce intestinal inflammation because it reversed the effects of the HFD on the distorted colonic crypt length, which is commonly observed in intestinal inflammation [44]. However, further studies are needed to understand why XOS on the LFD caused injuries similar to the HFD. This is surprising because XOS is known to be safe and well tolerated by normal human populations.

Of the prebiotic fibers, alpha-galacto-oligosaccharides [45] and fructo-oligosaccharides [46] have been shown to ameliorate NAFLD in preclinical animal models, but to the best of our knowledge, this study is the first to describe such an effect for XOS. Concomitantly with the increase in *F. prausnitzii*, XOS decreased the content of hepatic triglycerides in HFD-fed rats. This was explained by the XOS enhancing the activity of hepatic β-HAD, mitochondrial respiratory capacity available for the production of ATP through the electron flow from complex I, coupling of electron transport through the complex I and oxidative phosphorylation. β-HAD is a subunit of the mitochondrial trifunctional enzyme subunit alpha (MTP), which catalyzes the last three reactions of fatty acid β-oxidation. Previously, heterozygous MTP^+/−^ mice were shown to develop NAFLD simultaneously with a reduced rate of β-oxidation [47], thus highlighting the importance of MTP in the onset of the disease. Despite the improvements in hepatic metabolism, diet or XOS caused only small effects on the whole-body energy expenditure and respiratory quotient. However, as expected, the HFD enhanced the caloric intake, body weight gain and body fat content compared to the LFD. On the HFD, XOS diminished the levels of cecal metabolites tyrosine and isovalerate, whose reduction likely contributed to the increased hepatic fat oxidation. We assume this because higher levels of tyrosine have been observed in NASH patients [48] and, in addition, dysregulated metabolism of tyrosine has been documented in them [49]. The consequences of the dysregulated metabolism of tyrosine are not entirely clear, but it has been proposed that tyrosine could enter the ketogenic pathway, in which it will be metabolized directly to acetyl-CoA [50]. Therefore, high levels of tyrosine might enhance the synthesis of fatty acids and thus further contribute to the deposition of lipids in liver. The GM-produced isovalerate could also have participated in the NAFLD of the rats. This view is supported by a study in which the GMs from NAFLD mice and healthy mice were transplanted into germ-free mice. The mice receiving the GM of the NAFLD mice had significantly higher fecal isovalerate levels and reproduced NAFLD [51]. In our study, fecal nicotinic acid, butyrate and 2-oxoglutarate were associated negatively with hepatic fat content and positively with oxidative phosphorylation. Behind these associations may be the capacity of nicotinic acid to inhibit the flux of fatty acids from the adipose tissue to the liver [52]. Similarly, 2-oxoglutarate and butyrate are known to promote hepatic oxidative metabolism [53]. 

## 5. Conclusions

Our study provides evidence that specific, prebiotic dietary supplements can be used to ameliorate NAFLD. Further, we identified the enhanced hepatic oxidative metabolism and mitochondrial functions as the underlying prebiotic-dependent preventive mechanisms of NAFLD that further may be associated with the gut microbiota. However, our study was done in rats and it should be further studied in human cohorts whether our findings on XOS improving NAFLD can be extended to humans.

## Figures and Tables

**Figure 1 nutrients-12-03225-f001:**
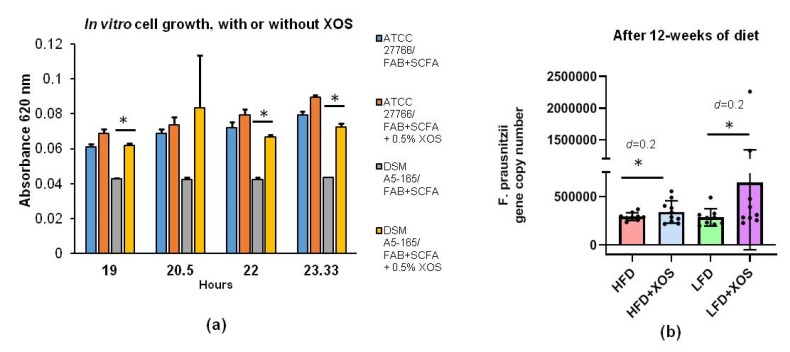
**Xylo-oligosaccharides** (XOS) increased the growth of *F. prausnitzii* in vitro and in vivo. (**a**) In vitro stimulation of *F. prausnitzii* growth with XOS. Cultivations of *F. prausnitzii* were done in fastidious anaerobe broth (FAB) supplemented with short-chain fatty acids (SCFA) in an anaerobic workstation. Ten mL cultures of commercially available *F. prausnitzii* strains American type of culture collections (ATCC)^®^-27766™ and Deutsche Sammlung von Mikroorganismen und Zellkulturen GmbH (DSM) A2-165 were supplemented or not with XOS as 0.5% (*w/v*%). The growth of *F. prausnitzii* was determined by following changes in the optical density at 620 nm. *n* = 4 replicates/treatments. * indicates statistically significant difference between the groups; (**b**) Quantitative real-time PCR of *F. prausnitzii* using DNA extracted from the rat cecum after 12 weeks of diet. The results are shown as gene copy number divided per gram of cecum content used to extract the total bacterial DNA. Cohen’s *d* was used to estimate the differences between the groups. *n* = 8–10/diet group. The different data points are shown with black dots. HFD = high-fat diet. LFD = low-fat diet. * indicates statistically significant difference between the groups.

**Figure 2 nutrients-12-03225-f002:**
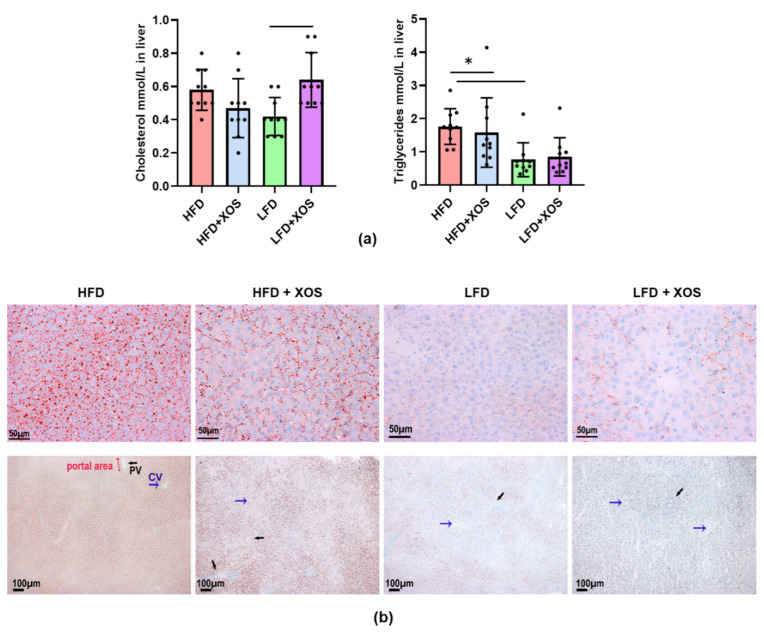
XOS decreased hepatic fat content in rats fed with HFD. (**a**) The biochemical analyses of the total hepatic cholesterol and triglyceride content. *n* = 8–10/diet group. The different data points are shown with black dots. * indicates statistically significant difference between the groups; (**b**) The frozen liver sections of the rats stained with Oil Red O. The scale bar in the upper images is 50 µM and in the lower images 100 µM. The histological images were taken with Olympus BX50 microscope and Colorview III camera using 40× (Olympus UPlanFI, NA 0.75) and 10× (Olympus UPlanFI NA 0.3) objectives. Blue arrows indicate central vein, which is abbreviated as CV in the lower image of the HFD. Black arrows indicate portal vein, which is abbreviated as PV in the lower image of the HFD.

**Figure 3 nutrients-12-03225-f003:**
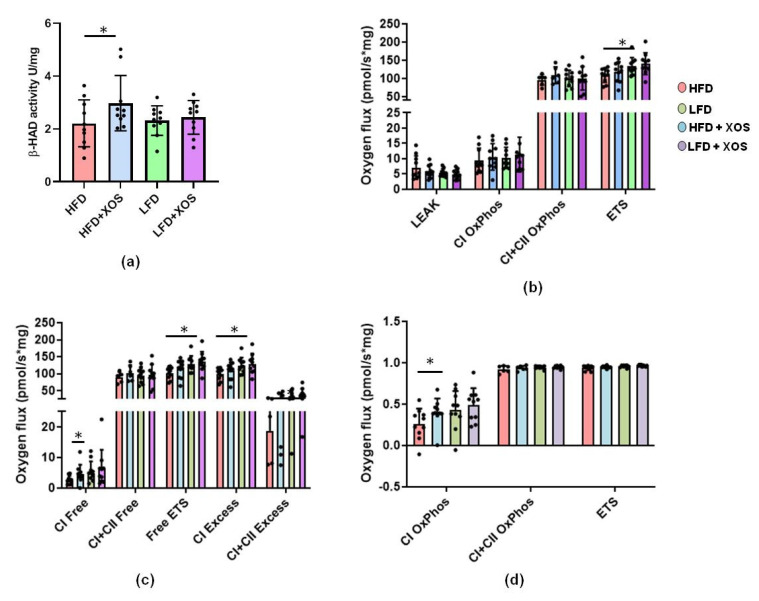
XOS enhanced the hepatic activity of fatty acid-oxidating beta-hydroxydeacetylase dehydrogenase (β-HAD) on the HFD and also increased mitochondrial respiration, reflecting increased glucose metabolism. (**a**) The biochemically measured activities of β-HAD in rat livers. *n* = 8–10/diet group; (**b**) HFD lowered maximal electron transport (ETS) in liver mitochondria. *n* = 8–10/diet group; (**c**) HFD lowered maximal electron transport capacity available for oxidative phosphorylation (Free ETS), reserve electron transport capacity beyond oxidative phosphorylation through complex I (CI Excess) as well as through complexes I and II (C+CII Excess). On the HFD, XOS increased the respiratory capacity available for the production of ATP through the electron flow from complex I (CI Free). *n* = 8–10/diet group; (**d**) On the HFD, XOS improved coupling of electron transport through the complex I and oxidative phosphorylation (CI OxPhos, coupling efficiency). *n* = 8–10/diet group. In all graphs, the different data points are shown with black dots. * indicates statistically significant difference between the groups in all graphs.

**Figure 4 nutrients-12-03225-f004:**
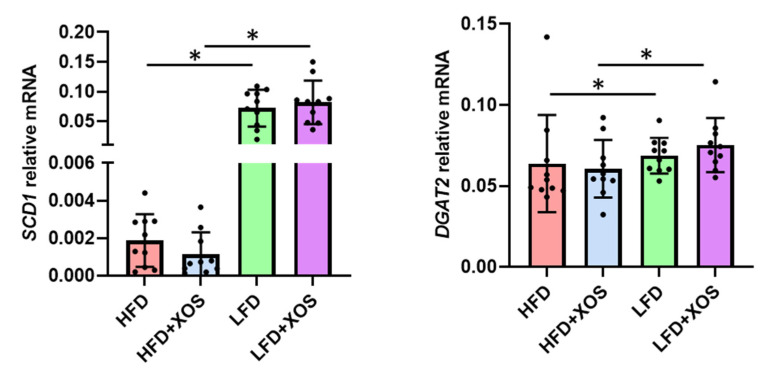
XOS did not affect the hepatic mRNA expression of stearoyl coenzyme desaturase 1 (SCD1) or Diacylglycerol O-acyltransferase 2 (DGAT2). *n* = 8–10/diet group. The different data points are shown with black dots. * indicates statistically significant difference between the groups.

**Figure 5 nutrients-12-03225-f005:**
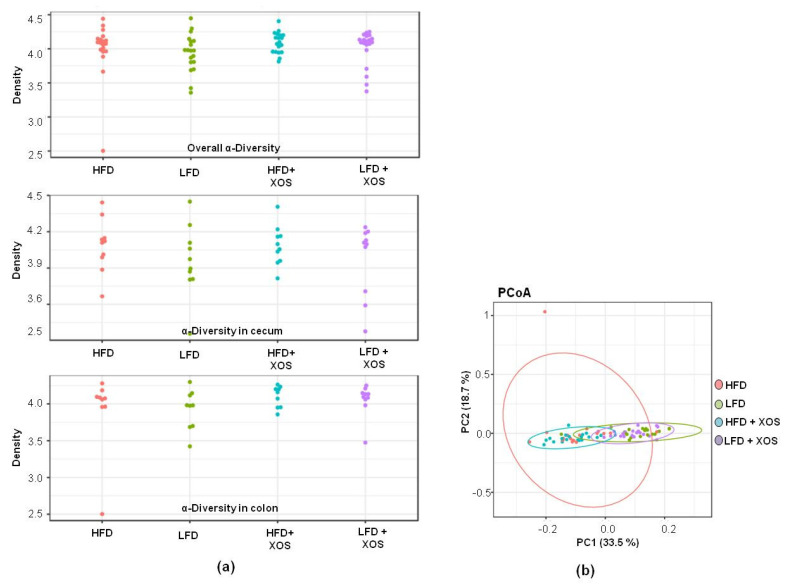
No significant associations were observed between the diets and the diversity of the gut microbiota. (**a**) The overall alpha-diversity (on top), alpha-diversity in colon (at the middle) and cecum (on bottom) of the gut microbiota. *n* = 10/diet group. The alpha-diversity of the gut microbiota was quantified with Shannon index using microbiome R/Bioconductor package; (**b**) The beta-diversity of the gut microbiota according to the principal component analysis (PCoA). PC1 indicates principal component 1 and PC2 principal component 2. *n* = 10/diet group. The analysis of beta-diversity was based on Bray–Curtis distance, and PERMANOVA was used for significance testing (adonis function in the vegan R package).

**Figure 6 nutrients-12-03225-f006:**
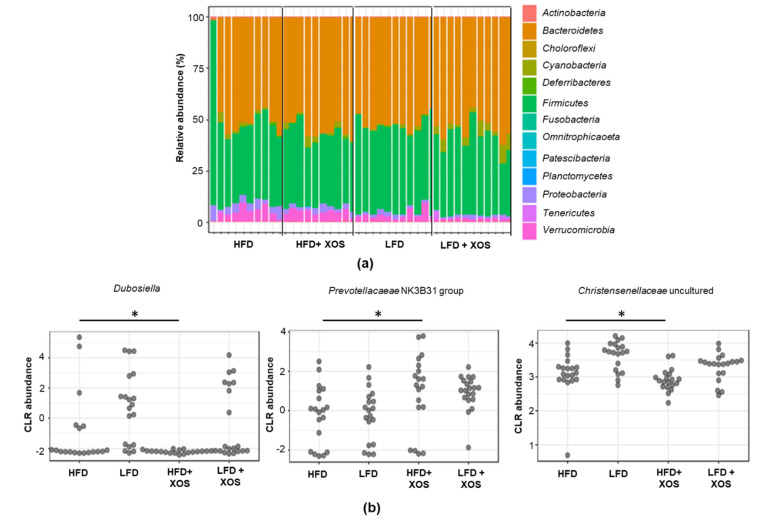
Differences were found in the abundance of three gut microbiota genera between the diet groups. (**a**) The gut microbiota composition of the rats at phylum level did not differ between the groups. *n* = 10/diet group. The taxonomic groups were compared based on DESeq2 using phyloseq R/Bioconductor package including Benjamini–Hochberg correction for multiple testing. (**b**) Differences between the groups were found in the relative abundance of *Dubosiella*, Prevotellaceae NK3B31 and uncultured genus of Christensenellaceae family. *n* = 10/diet group. The taxonomic groups were compared based on DESeq2 using phyloseq R/Bioconductor package including Benjamini–Hochberg correction for multiple testing. CLR abundance = centered log-ratio transformed abundance. * indicates statistically significant difference between the groups.

**Figure 7 nutrients-12-03225-f007:**
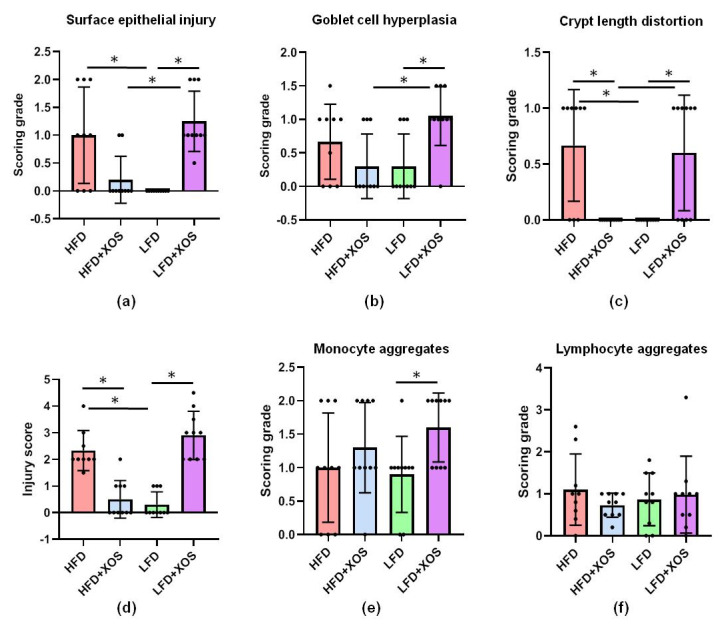
XOS improved the HFD-induced injury and inflammatory markers in the proximal colon. The histopathological examination detected an interactive effect of diet and XOS on the (**a**) surface epithelial injury; (**b**) Goblet cell hyperplasia; (**c**) crypt length distortion and; (**d**) injury score; (**e**) On the LFD, XOS increased monocyte aggregates; (**f**) The diets did not affect the lymphocyte aggregates. *n* = 8–10/diet group. The different data points are shown with black dots in all graphs. * denotes statistically significant difference between the groups in all graphs.

**Figure 8 nutrients-12-03225-f008:**
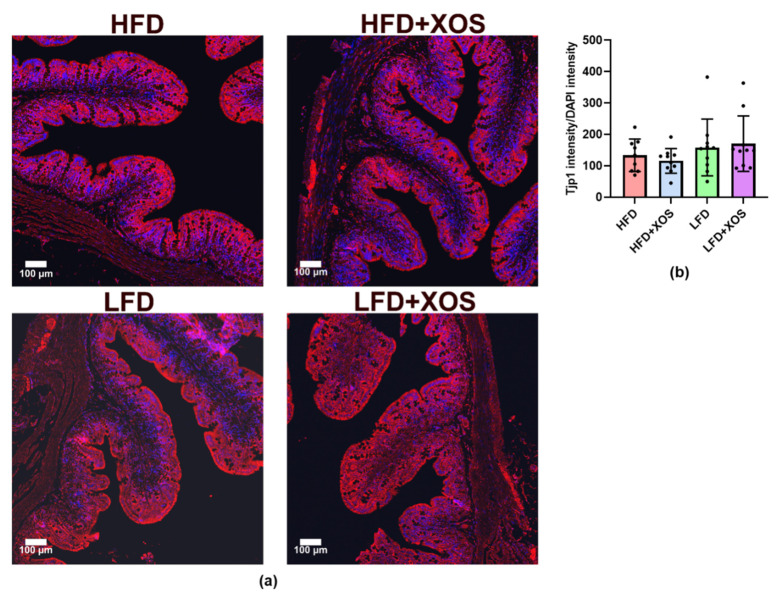
The intestinal Tjp1 did not differ between the diet groups. (**a**) The histological images were taken with Zeiss laser scanning microscopy (LSM) 700 and 20 x Plan-Apochromat 20 x/0.8 M27 objective. Tjp1 is shown with red label and 4′,6-diamidino-2-phenylindole (DAPI) in blue. The scale bar is 100 µM; (**b**) The bars in the graph represent the expression of Tjp1 counted as its intensity using Image J adjusted to the intensity of DAPI. *n* = 9–10/diet group. The different data points are shown with black dots.

**Figure 9 nutrients-12-03225-f009:**
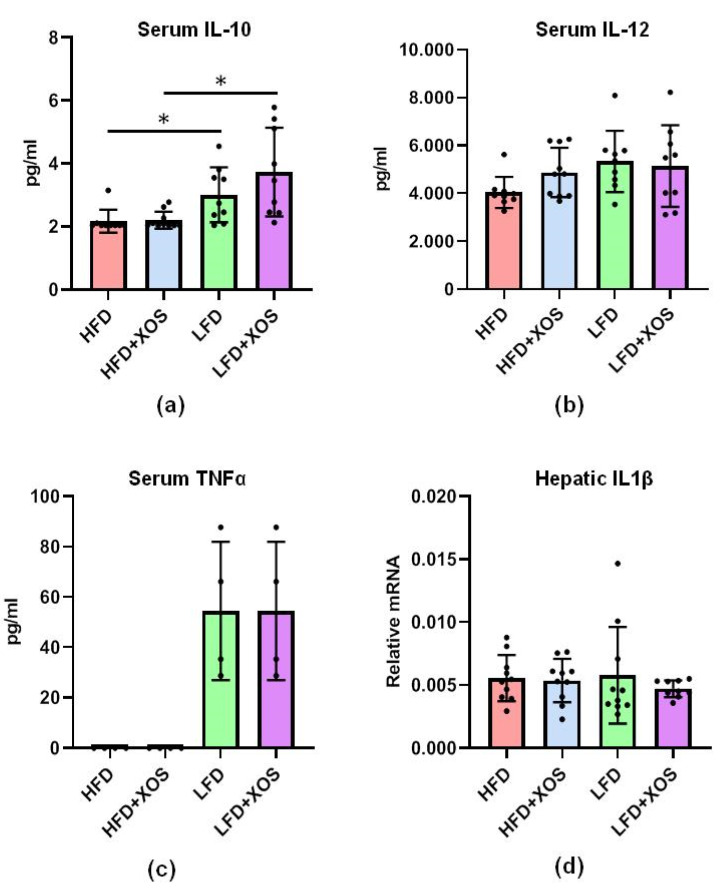
XOS supplementation did not affect the serum levels of (**a**) IL-10; (**b**) IL-12 or; (**c**) TNFα or; (**d**) the hepatic IL1β mRNA but the LFD groups had higher levels of anti-inflammatory IL-10. The serum cytokines were analyzed after 12 weeks of diet intervention using 9-plex ELISA, Quansys and Q-View software. Out of nine cytokines, only the levels of IL-10, IL-12 and TNFα were detectable in rats. IL1β mRNA was analyzed from hepatic cDNA with qPCR. *n* = 8–10/diet group. The different data points are shown with black dots in all graphs. * indicates statistically significant difference between the groups in all graphs.

**Figure 10 nutrients-12-03225-f010:**
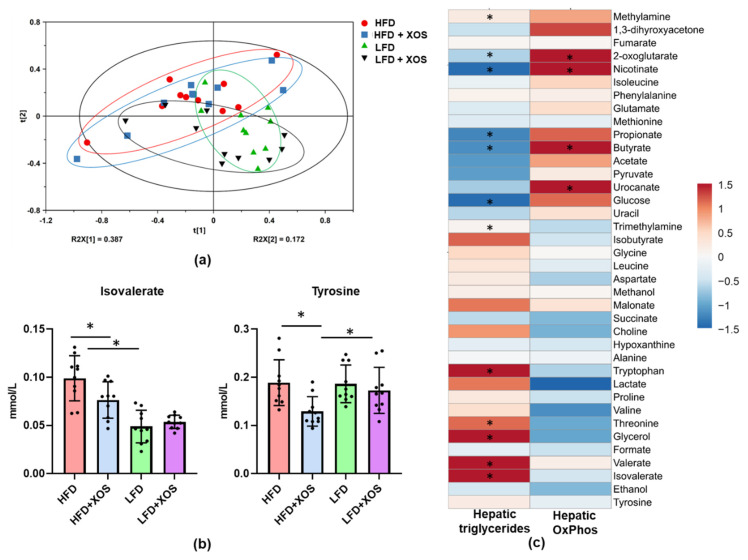
XOS decreased the levels of cecal tyrosine and isovalerate on the HFD. (**a**) The score scatter plot of the Principal Component Analysis (PCA) of the cecal metabolites showed no major differences between the diet groups. *n* = 10/diet group; (**b**) The levels of cecal tyrosine and isovalerate differed between the diet groups. *n* = 10/diet group. The different data points are shown with black dots. * indicates significant difference between the groups; (**c**) The associations of the metabolites with the hepatic triglyceride content and oxidative phosphorylation. *n* = 10/diet group. The heatmap was drawn using Clustvis, which is a web tool used to visualize clustering of multivariate data (https://biit.cs.ut.ee/clustvis/). However, the clustering is not shown here. * indicates significant association between the variables. The colored scale bar for the associations is shown on the right, and the color corresponds to the value of Spearman’s correlation coefficient.

**Figure 11 nutrients-12-03225-f011:**
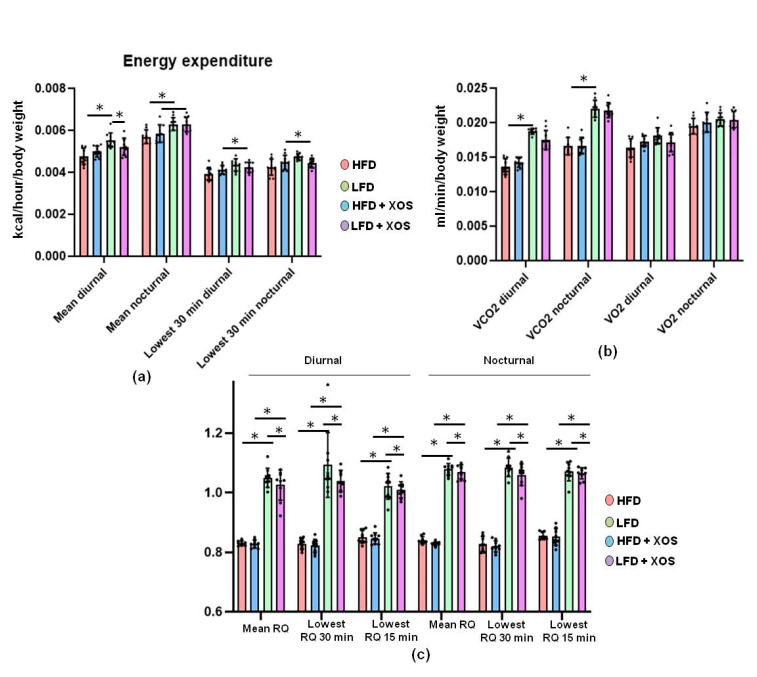
The LFD increased the energy expenditure in rats, while XOS and diet had an interactive effect on the respiratory quotient. (**a**) The average hourly energy expenditure was highest at nighttime in the LFD and the lowest measured energy expenditure was highest in the LFD at daytime; (**b**) The HFD diminished the production of CO_2_ at daytime, whereas in O_2_ consumption, no differences between the groups were found; (**c**) The respiratory quotient (RQ) values were calculated as VCO_2_/VO_2_. The different data points are shown with black dots. * indicates statistically significant difference between the groups.

**Figure 12 nutrients-12-03225-f012:**
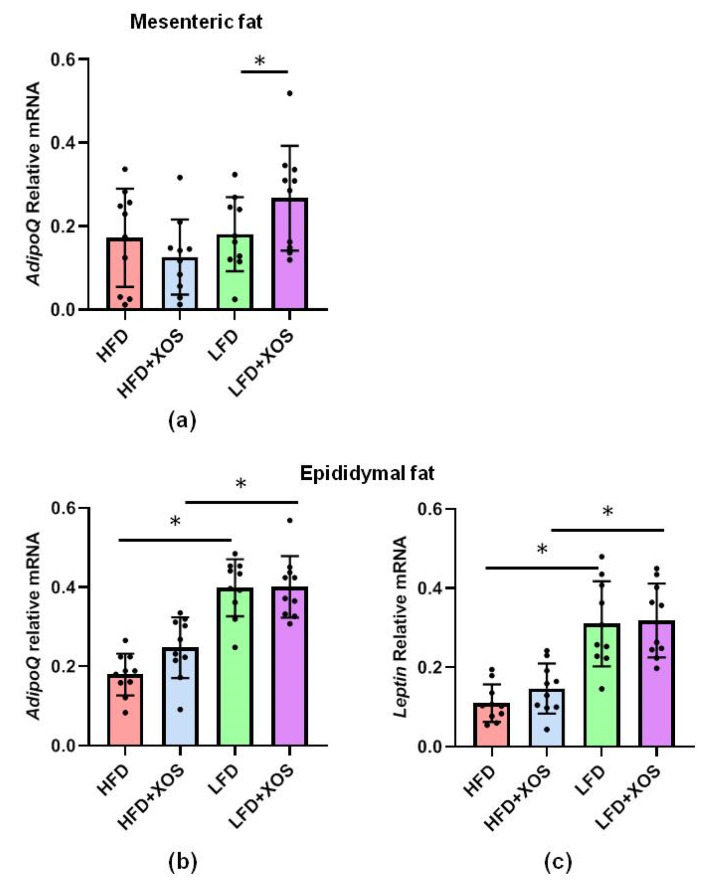
On the HFD, XOS did not affect the expression of leptin or adiponectin mRNA. The diet or XOS had no main effects on the expression of (**a**) adiponectin (AdipoQ) mRNA in the mesenteric adipose tissue, but on the LFD, XOS had an increasing effect on its expression. In the epididymal adipose tissue the diet but not XOS had an effect the expression of (**b**) leptin and; (**c**) AdipoQ mRNA. *n* = 8–10/diet group. The different data points are shown with black dots in all graphs. * indicates statistically significant difference between the groups in all graphs.

**Table 1 nutrients-12-03225-t001:** The concentrations of SCFA (mM, mean ± SEM) and acetate (A): propionate (P): butyrate (B) ratio in the cecum of the rats.

Group (*n* = 10/Group)	Acetate (A)	Propionate (P)	Butyrate (B)	Ratio A:P:B
HFD	16 ± 1.2	3.4 ± 0.22	0.91 ± 0.16	79:16:5
HFD+XOS	16 ± 1.4	3.4 ± 0.29	0.96 ± 0.13	79:16:5
LFD	22 ± 0.99	5.2 ± 0.22	2.6 ± 0.39	74:17:9
LFD+XOS	20 ± 1.2	4.9 ± 0.26	3.0 ± 0.41	72:17:11

SCFA, short-chained fatty acids; SEM, standard error of means; XOS, xylo-oligosaccharides HFD, high-fat diet, LFD, low-fat diet.

**Table 2 nutrients-12-03225-t002:** Serum clinical variables after the 12 weeks of diet intervention. The values are presented as mean ± standard deviation (SD). *n* = 8–10/diet group.

Serum Variable	HFD Mean ± SD	HFD+XOS Mean ± SD	LFD Mean ± SD	HFD+XOS Mean ± SD
Trigly (mmol/L)	2.50 ± 0.59	3.00 ± 0.95	2.27 ± 0.84	2.68 ± 0.74
FFA (µmol/L)	351 ± 158	370 ± 127	281 ± 153	288 ± 132
Chol (mmol/L)	3.04 ± 0.25	2.85 ± 0.31	2.80 ± 0.47	2.84 ± 0.59
LDL (mmol/L)	0.52 ± 0.17	0.42 ± 0.20	0.45 ± 0.16	0.46 ± 0.19
HDL (mmol/L)	1.76 ± 0.16	1.60 ± 0.41	1.81 ± 0.38	1.62 ± 0.35
Glycerol (µmol/L)	292 ± 59	313 ± 62	190 ± 40	196 ± 64
ALT (U/L)	35.90 ± 4.53	37.20 ± 4.52	22.78 ± 4.12	21.60 ± 3.20
AST (U/L)	68.10 ± 5.36	71.67 ± 7.68	65.90 ± 12.62	57.89 ± 10.42
Insulin (iU/L)	2.50 ± 0.95	2.00 ± 0.51	2.30 ± 0.70	2.17 ± 0.64
Glucose (mmol/L)	14.30 ± 2.79	14.85 ± 2.49	13.17 ± 3.31	12.04 ± 2.13

Abbreviations: Trigly, triglycerides; FFA, free fatty acids; Chol, total cholesterol; LDL, low-density lipoprotein cholesterol; HDL, high-density lipoprotein cholesterol; AST, aspartate aminotransferase; ALT, alanine aminotransferase; SD, standard deviation.

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
