# Peer review of "Prebiotic Xylo-Oligosaccharides Ameliorate High-Fat-Diet-Induced Hepatic Steatosis in Rats"

_nutrients, 2020, doi:10.3390/nu12113225_

Round 1
Reviewer 1 Report
(Remarks to the Author):
Nonalcoholic fatty liver disease (NAFLD) has been reported to be associated with altered intestinal bacterial communities and is a highly prevalent comorbidity of many chronic diseases. Based on this association, researches focus on alternative treatments aiming at modulating intestinal microbiota has prompted. The use of prebiotics to enhance and improve the balance of commensal bacteria in the gut is not new, however, scarce evidence regarding their effectiveness in clinical populations exists. The manuscript by Lensu and colleagues investigate the role of prebiotic xylo-oligosaccharides (XOS) in facilitating F. prausnitzii growth and improving high fat diet (HFD)-induced abnormality including reduction of caecal isovalerate and tyrosine in vitro and in vivo. The involvement of the gut microbiota (GM) in NAFLD has led to the development of possible therapies. Overall, the experiments are performed well, statistics are appropriate and the writing is solid. The findings will influence thinking in the field however I think that significant caveats to the model system used should be highlighted not to overstate the certainty of this process in the human. There are a number of concerns that must be addressed, and these are detailed as follows.
Central limitations of this study are:
1. The effect of XOS on F. prausnitzii growth was assessed in vitro. One time point was selected for bacteria concentration examination. Why 22h time point was selected? Time-course examination, at least three time points, should be performed to clearly understand the effect of XOS on the growth of F. prausnitzii in vitro. The same experiment should be applied to Fig. 1b and c, long-term effect of XOS needs to be address especially.
2. This is the first study to display the role of XOS in F. prausnitzii, the dose-dependent effect of XOS should be addressed.
3. Biomarkers offer a potential prognostic or diagnostic indicator for disease manifestation or progression. Serum biomarkers of NAFLD, such as total cholesterol, triglycerides, insulin resistance, leptin, adiponectin, and C-peptide, have been reported for many years. Authors should include these typical NAFLD markers to get more detail information of the effect of XOS on the progression of NAFLD.
4. The Oil Red O staining is not consistent with the result from cholesterol and triglycerides between LFD and LFD+XOS group. Why LFD group shows no Oil Red signal? Additionally, lipogenesis is enriched pericentral lobule, zonal distribution should be outlined.
5. How is the fibrosis state of the liver before and after XOS treatment?
6. High rates of hepatic lipid oxidation and lipogenesis are central features of NAFLD in both rodents and humans, and lipolysis and lipogenesis are critical for lipid accumulation in the liver. Therefore, not only fat oxidation but also lipogenesis need to be examined.
7. In Fig. 5, the quality of images needs improvement and high-power field should be displayed.
8. HFD significantly increased paracellular permeability in both the small and large intestine has bee demonstrated. The authors showed that Tjp1, also called ZO1, has no significant change, which is controversial to previous studies. An explanation is needed. Also, more markers need to include in Fig. 5, to demonstrate that XOS indeed improve HFD-caused abnormality.
9. The information of each individual sample should be pointed out in the bar graphs, if the numbers of sample are less ten (including ten).

Author Response
RESPONSES TO THE COMMENTS AND CONCERNS OF REVIEWER 1
- The effect of XOS on F. prausnitzii growth was assessed in vitro. One time point was selected for bacteria concentration examination. Why 22h time point was selected? Time-course examination, at least three time points, should be performed to clearly understand the effect of XOS on the growth of F. prausnitzii in vitro. The same experiment should be applied to Fig. 1b and c, long-term effect of XOS needs to be address especially.
Our response: For clarity, we had selected only one time point for the figure. However, the growth was also assessed after 19, 20.5 and 23.33 hours. As requested, we have now replaced the Figure 1a by a new figure showing the growth in all time points that were studied.
These changes can be seen in Figure 1a, page 9, and in the text lines 389-392, page 9.
As to the comment on Figures 1b and c, for simplicity we chose only one time point here. We agree with the reviewer in that chronic effects of XOS have not been reported, but sub-chronic effects on the microbiota have been shown already previously following 6-month intervention (Santos et al. 2006, https://doi.org/10.1016/j.fm.2005.07.004). The effects of XOS have been also evaluated in mice following high 32 g/kg dose, and in rats with dietary administration at concentrations of 0 (control), 0.9, 2.9, 8.8 and 10% for 13 weeks (DOI: 10.3109/15376516.2012.706837).
- This is the first study to display the role of XOS in F. prausnitzii, the dose-dependent effect of XOS should be addressed.
Our response: We agree with the reviewer and understand this comment from the pharmacological point of view. However, in this study we did not have resources for the dose-response determination, and the study relies on in vitro data where we found an effective concentration for the F. prausnitzii growth. Based on that and the existing human data [27, Finegold et al. 2014], we chose a dose that would be prebiotic. An interesting thing with XOS is that compared to other commercially available prebiotics, it confers other health benefits to the host with much lower concentrations. For humans, the current recommendations of daily dose of XOS vary between 1 and 2.8 grams. The dose that we used for rats is comparable to that for humans. XOS was approved for human use in Europe in 2017.
We have included information on this issue in lines 132-135, page 3.
- Biomarkers offer a potential prognostic or diagnostic indicator for disease manifestation or progression. Serum biomarkers of NAFLD, such as total cholesterol, triglycerides, insulin resistance, leptin, adiponectin, and C-peptide, have been reported for many years. Authors should include these typical NAFLD markers to get more detail information of the effect of XOS on the progression of NAFLD.
Our response: We thank the reviewer for this valuable comment. We apologize that the reviewer may have missed that most of the serum biomarkers were included in the initial version of the manuscript because they were placed in the supplementary file. We have now removed them from the supplementary material and instead have included a table of the results within the main text. Unfortunately, because of the short supply of the serum sample, we were unable to perform new analysis on leptin, adiponectin, and C-peptide that were requested. Therefore, we have added analyses of the expression of adiponectin and leptin in in the mesenteric and epididymal adipose tissues. The diet or XOS had no main effects on the expression of adiponectin (AdipoQ) mRNA in the mesenteric adipose tissue, but on the LFD, XOS had increasing effect on its’ expression [F (1,34) = 4.6, P = 0.04]. In the epididymal adipose tissue, the diet but not XOS had an effect the expression of leptin [F (1,36 ) = 25.5, p <0.001] and AdipoQ [F (1, 36) = 8.9, p = 0.005] mRNA, HFD diminishing the expression of both.
The table of the markers can be found in page 19. The results of the requested new analyses on leptin and adiponectin can be found in figure 10, page 20, and the corresponding text in page 19.
- The Oil Red O staining is not consistent with the result from cholesterol and triglycerides between LFD and LFD+XOS group. Why LFD group shows no Oil Red signal? Additionally, lipogenesis is enriched pericentral lobule, zonal distribution should be outlined.
Our response: We apologize for the confounding figure of the LFD group. We have now replaced it with a more representative figure. We have also added figures with lower and higher magnification so that the distribution can be better estimated.
The new figure can be found in page in page 10.
- How is the fibrosis state of the liver before and after XOS treatment?
Our response: To observe fibrosis, the liver sections were stained with Sirius Red after the intervention in all groups. Examples of the typical stainings are now shown as a Supplement, Figure S8. Some amount of fibrosis was detected but this variable seemed equal among the groups and therefore we did not include the data or further analyses in the original submission of the manuscript.
The new results can be found in lines 422-424, page 9, and the new figure S1 in supplementary material.
- High rates of hepatic lipid oxidation and lipogenesis are central features of NAFLD in both rodents and humans, and lipolysis and lipogenesis are critical for lipid accumulation in the liver. Therefore, not only fat oxidation but also lipogenesis need to be examined.
Our response: This is a good point indeed. We have added results on the mRNA expression of SCD1 and DGAT2 as markers of hepatic lipid oxidation and lipogenesis. The LFD groups had higher expression of SCD1, which is in accordance with previous reports by others, e.g. https://www.ncbi.nlm.nih.gov/pmc/articles/PMC3786952/
The new results can be found in lines 466-468, page 11 and the new figure in page 12.
- In Fig. 5, the quality of images needs improvement and high-power field should be displayed.
Our response: Please see the answer for the question 8 below.
- HFD significantly increased paracellular permeability in both the small and large intestine has bee demonstrated. The authors showed that Tjp1, also called ZO1, has no significant change, which is controversial to previous studies. An explanation is needed. Also, more markers need to include in Fig. 5, to demonstrate that XOS indeed improve HFD-caused abnormality.
- The information of each individual sample should be pointed out in the bar graphs, if the numbers of sample are less ten (including ten).
Our response: We thank the reviewer for this notion. We have included the information to the figure legends.
Reviewer 2 Report
Sanna Lensu and colleagues performed a study with the aim to identifies F. prausnitzii as a possible target to treat NAFLD with XOS.
The topic of the study is interesting since Faecalibacterium prausnitzii is a commensal member of the gut microbiota with known anti-inflammatory properties and it is known that administration of F. prausnitzii prevented NAFLD in mice.
However, it presents some points that need to be addressed, as following:
Minor Points
- The Results section is not well structured and it is difficult to understand due to the numerous subsections. I would like to see a shorten, more readable version of this section.
- Several studies reported the anti-inflammatory properties of Faecalibacterium prausnitzii . However in this study XOS supplementation and thus higher abundance of F. prausnitzii did not affect the levels of IL-10, IL-12 and TNFα. Please comment in discussion section upon how inflammatory cytokines were not influenced by higher abundance of F. prausnitzii. Why did the authors not evaluate the levels of adiponectin which has been extensively studied for its anti-inflammatory effects in NAFLD patients?
- The discussion section is excessively long—it can be made more succinct and to the point.
- Please add captions to table 1
- Numerous typographical and grammatic errors have to be corrected.
Author Response
RESPONSES TO THE COMMENTS AND CONCERNS OF REVIEWER 2
- The Results section is not well structured and it is difficult to understand due to the numerous subsections. I would like to see a shorten, more readable version of this section.
Our response: We thank the reviewer for the suggestion. We have done our best to improve the text of the results section within the time that was given for the modifications. We hope that the corrections satisfy the reviewer.
- Several studies reported the anti-inflammatory properties of Faecalibacterium prausnitzii . However in this study XOS supplementation and thus higher abundance of F. prausnitzii did not affect the levels of IL-10, IL-12 and TNFα. Please comment in discussion section upon how inflammatory cytokines were not influenced by higher abundance of F. prausnitzii. Why did the authors not evaluate the levels of adiponectin which has been extensively studied for its anti-inflammatory effects in NAFLD patients?
Our response: It is well known that the levels of cytokines and their production are not constant variables. Therefore, it should be noted that we were looking the levels of the cytokines only at the end point, and thus may have missed whether their concentrations have been elevated in some other time point during the 12-weeks diet intervention. This limitation has now been added to the discussion.
Unfortunately, because of the short supply of the serum sample, we were unable to perform new analysis on serum adiponectin. Therefore, we have added analyses of the expression of adiponectin in adipose tissues. The diet or XOS had no main effects on the expression of adiponectin (AdipoQ) mRNA in the mesenteric adipose tissue, but on the LFD, XOS had increasing effect on its’ expression [F (1,34) = 4.6, P = 0.04]. In the epididymal adipose tissue, the diet but not XOS had an effect the expression of leptin [F (1,36 ) = 25.5, p <0.001] and AdipoQ [F (1, 36) = 8.9, p = 0.005] mRNA, HFD diminishing the expression of both.
The results of the requested new analyses on leptin and adiponectin can be found in figure 10, page 20, and the corresponding text in page 19.
- The discussion section is excessively long—it can be made more succinct and to the point.
Our response: We agree and thank the reviewer for this suggestion. We have made our best to shorten the discussion, and we hope that it is now more succinct.
- Please add captions to table 1
Our response: There is a caption in table 1.
- Numerous typographical and grammatic errors have to be corrected.
Our response: Language revisions as well as correction of typographical and grammatic errors have been made.
Reviewer 3 Report
This study examines the efficacy of xylo-oligosaccharides as a prebiotic on NAFLD in rats. The authors show that xylo-oligosaccharides reduce triglycerides in the liver in the rats with NAFLD and that the mechanism is due to enhanced hepatic mitochondrial function and enhanced oxidative metabolism. Aslo, they show that XOS increases the Fecalibacterium prausnitzi in rats' colon.
The mechanisms by which prebiotics and gut microbiota ameliorate NAFLD are not fully clarified, and this study is very interesting. However, it would become a more important report if the following points were revised.
・This report shows that XOS increases Fecalibacterium and that XOS improves NAFLD. However, whether the effects of XOS on NAFLD are mediated by gut bacteria, including Fecalibacterium, have not been investigated. The title should also be revised as it seems to indicate that XOS ameliorate NAFLD via Fecalibacterium.
・Liver fibrosis has not been assessed; fibrosis should also be assessed because the most important prognostic factor in NAFLD is fibrosis. The authors may need to use other NAFLD models with liver fibrosis as well.
・With regard to intestinal permeability, only TJP1 is being examined. It is difficult to determine changes in intestinal permeability based on changes in TJ1 expression alone. Other tight junction-related proteins should also be examined. Also, intestinal permeability should be examined by methods other than protein expression.
・Inflammatory cytokines in the blood were evaluated. Systemic inflammation is less important in this study and it would be better to evaluate inflammatory cytokines in the gut and liver.
・Cristophersen et al. reported that XOS reduced Fecalibacterium. The difference in results from this study should be discussed.
Author Response
RESPONSES TO THE COMMENTS AND CONCERNS OF REVIEWER 3
This report shows that XOS increases Fecalibacterium and that XOS improves NAFLD. However, whether the effects of XOS on NAFLD are mediated by gut bacteria, including Fecalibacterium, have not been investigated. The title should also be revised as it seems to indicate that XOS ameliorate NAFLD via Fecalibacterium.
Our response: We agree with this comment and have removed Faecalibacterium prausnitzii from the title. We have also modified our conclusions in a way that they do not state that we studied mediation by gut bacteria.
The modified conclusions can be found in the abstract, lines 35-38, and in the discussion lines 705-712, page 22.
・Liver fibrosis has not been assessed; fibrosis should also be assessed because the most important prognostic factor in NAFLD is fibrosis. The authors may need to use other NAFLD models with liver fibrosis as well.
Our response: To observe fibrosis, the liver sections were stained with Sirius Red after the intervention in all groups. Examples of the typical stainings are now shown in Supplementary figure S1. A minimal amount of fibrosis was detected but this variable seemed equal among the groups and therefore we did not include the data or further analyses in the original submission of the manuscript.
The new results can be found in lines 422-424, page 9, and the new figure S1 in supplementary material.
・With regard to intestinal permeability, only TJP1 is being examined. It is difficult to determine changes in intestinal permeability based on changes in TJ1 expression alone. Other tight junction-related proteins should also be examined. Also, intestinal permeability should be examined by methods other than protein expression.
Our response: Unfortunately, within the time limits we were unable to analyze histologically many more tight junction-related proteins and unfortunately, we are also lacking material for other methods such as qPCR or Western blot. However, we have included more intestinal markers to show the HFD-caused abnormalities that were improved by XOS supplementation. We hope that these new results satisfy the reviewer. In the histopathological assessment, we found that in the HFD group there were more epithelial injuries in the proximal colon resulting in a higher injury score. In addition, the crypts in the proximal colon had distorted length. On the HFD, XOS supplementation reversed these effects of HFD on the gut. The scoring values for HFD+XOS were similar to the values of LFD. However, alarmingly XOS supplementation on the HFD caused injuries similar to the HFD.
Regarding Tjp1 analyses, after a more detailed inspection of the microscopy data we have now realized that the data should be re-analyzed. It seems that the fluorescence intensity was too high and also epithelial cytoplasm has been stained that compromises the analysis. Because we could not re-analyze the data within one week, we have excluded the Tjp1 data from the manuscript.
The new results can be found in lines 519-526, page 14, and the figure in page 15. In addition, the results are discussed in lines 631-634 and lines 657-661, page 21.
・Inflammatory cytokines in the blood were evaluated. Systemic inflammation is less important in this study and it would be better to evaluate inflammatory cytokines in the gut and liver.
Our response: We agree with that the inflammation in liver is important and have therefore now analyzed the mRNA expression of IL1β in liver. In addition, monocyte and lymphocyte aggregates have been determined from proximal colon samples and the data is now included in the manuscript. Distorted crypth length has been previously shown to associate with HFD-caused intestinal inflammation and is one of the most prevalent markers of IBD. Thus likely, it serves also in our study as a marker of intestinal inflammation.
The hepatic IL1β results can be found in figure 8, page 16, as well as in the text lines 538-540, page 15.
・Cristophersen et al. reported that XOS reduced Fecalibacterium. The difference in results from this study should be discussed.
Our response: We thank the reviewer for this valuable comment because there are indeed important differences between the two studies. We have now added to the discussion: “Christophersen et al. found that XOS decreased the growth of F. prausnitzii [28]. However, there are methodological differences between the latter and our study that may explain the controversies. Christophersen et al. first stimulated the GM with 3% soy protein to study its’ detrimental effects on gut health. Then, the treatment was followed by soy protein with 1% XOS. Thus, it may be that after the decreasing effects of soy bean on the abundance of F. prausnitzii, the XOS may not be sufficient to increase its’ abundance”.
This discussion can be found in lines 622-625, page 20.
Round 2
Reviewer 1 Report
Remarks to the Authors
I am grateful for the additional work provided by the authors in the revision of the manuscript. The additional experiments are particularly welcome and are a significant addition. Overall these revisions have improved the manuscript significantly.
In response to my comments.
1. Two-way ANOVA with post-test need to be applied for examining the significance of each growth curve.
2. It is crucial that it is explicitly stated that the dose effect of XOS on cells growth. At least two dosages need to be used for checking the effect of XOS on cell growth.
3. I accept the author’s responses.
4. I accept the new Fig 2., but please indicate the location of central vein and portal triad in Fig 2b.
5. Examples of the fibrosis staining showed in Figure S1 needs to be quantified.
6. I agree with the changes made in response to my comment #6.
7 & 8. In the previous version, authors showed that no difference of Tjp1 between the diet groups, which is critical information for XOS treatment, although it was controversial to previous reports. The Figure 5 should be kept, and one more marker needs to be examined. If this phenomenon is real, the authors need to have discussion.
9. The presence of individual data points is necessary as sample numbers less than ten (including ten).
10. Further proof reading and corrections are required.
Author Response
Please find our point point reponses below.
- Two-way ANOVA with post-test need to be applied for examining the significance of each growth curve.
Our response: We thank the reviewer for this comment and we have now revised the statistics for the growth curves. Two-way ANOVA with post-test was used to compare the significance of each growth curve. In fact, we tested the effect of celline, time and XOS-treatment by three way analysis of variance, and the new results are now added to the text. Here are the statistics that have been added to the results section lines 423-427, page 9, and in addition the Figure 1a has been modified accordingly:
Compared to the control treatment, XOS increased the growth of F. prausnitzii in vitro [F (1, 16.9) = 64.9, p<0.001]. The effects of time [F (3, 25.6) = 5.4, p = 0.005] and cellline [F (1, 16.9) = 59.6, p < 0.001] were also significant. Thus, the difference between control and XOS treatment was significant in each time (after 19, 22, 23.3 h of stimulation) in the DSM A5-165 strain (p < 0.05) but not in the ATCC-27766 strain (Figure 1a).
- It is crucial that it is explicitly stated that the dose effect of XOS on cells growth. At least two dosages need to be used for checking the effect of XOS on cell growth.
Our response: We did some initial testing with different concentrations of XOS. For example, 0.2% of XOS also increased the growth. However, at that time we estimated the growth by plating and counting the F. prausnitzii cells. Therefore, we do not feel that the results are adequate enough to be added to the manuscript. Further, the original purpose of the in vitro study was actually just to show that F. prausnitzii can utilize XOS and therefore we did not aim to test actual dose responses. Unfortunately, due to the ongoing Covid-19 pandemic we are currently unable to perform complementary in vitro analyses at the given time frame, which is 5 days. Currently, the situation is very bad in Finland, especially where we live and all microbiological facilities are allocated to Covid-19 research and testing. Further, at the university we are only allowed to do remote work, which means that the only corrections that we can do the manuscript now are the ones that can be done with the computer. We are very sorry about this inconvenience and we hope that the reviewer understands this situation. During the first round of revision, we were able to provide the requested results because the experiments were done before the Covid-19 pandemic started.
To describe the purpose of the in vitro experiment better, we have modified in the introduction the lines 97-100, page 3 in a following way:
In the present study, we first tested in vitro whether F. prausnitzii can utilize XOS and how a 0.5% XOS affects the growth of F. prausnitzii, and then studied the effects of a prebiotic dose of XOS (0.12%) in vivo.
And in results lines 422-423, page 9 we have added the following sentence:
We first tested in vitro whether F. prausnitzii can utilize XOS and how XOS affects its’ growth.
- I accept the author’s responses.
Our response: We thank the reviewer by accepting our response.
- I accept the new Fig 2., but please indicate the location of central vein and portal triad in Fig 2b.
Our response: We have now added the missing information in the figure 2.
- Examples of the fibrosis staining showed in Figure S1 needs to be quantified.
Our response: We are sorry for the missing information. The quantification of fibrosis has now been done based on Ishak grade. Briefly, 0 is no fibrosis; 1 is fibrous expansion of some portal areas and short fibrous septa; 2 is fibrous expansion of most portal areas and short fibrous septa; 3 is fibrous expansion of most portal areas and occasional portal to portal bridging; 4 is fibrous expansion of portal areas and marked portal to portal as well as portal to central bridging; 5 is marked bridging and occasional nodules, and; 6 is probable or definite cirrhosis.
Based on the scoring results, we have added to the results section the following interpretation to lines 459-463, page 10:
Based on the Sirius Red staining, there was no pronounced progression to hepatic fibrosis in any of the diet groups or differences in the fibrosis score between the groups. Yet, there was at some extent fibrous expansion in portal areas and short fibrous septa in all groups, which could be due to having feed ad libitum. Examples of typical stainings are shown in Supplementary figure S1.
A graph of the fibrosis scoring results has been added to the Supplementary figure S1.
- I agree with the changes made in response to my comment #6.
Our response: We thank the reviewer by accepting our response.
7 & 8. In the previous version, authors showed that no difference of Tjp1 between the diet groups, which is critical information for XOS treatment, although it was controversial to previous reports. The Figure 5 should be kept, and one more marker needs to be examined. If this phenomenon is real, the authors need to have discussion.
Our response: We have again included the Tjp1 -figure and results in the manuscript. We agree with the reviewer that more markers could be included, however, please refer to our response above that currently we are unable to perform such massive laboratory analyses. Moreover, even without the Covid-19 situation it would impossible to do such extensive analyses of 40 rats within 5 days that were given for the revision. But definitely, those are worth of publishing in a future work whenever it will be possible to work again in the laboratory.
Therefore, we have added to the discussion lines 693-698, page 24 the following:
However, XOS did not affect intestinal Tjp1 according to our immunohistological analysis. The latter finding was somewhat surprising because we have earlier shown that F. prausnitzii administration increased the expression of intestinal Tjp1 mRNA [23], and others have shown that prebiotic oligosaccharides in general can prevent HFD-induced impairment of gut permeability [39]. Thus, further studies are needed to determine the effects of XOS on other tight junction proteins.
- The presence of individual data points is necessary as sample numbers less than ten (including ten).
Our response: We have modified all figures to show the individual data points in the graphs. We hope that the new graphs satisfy the reviewer.
- Further proof reading and corrections are required.
Our response: We have made our best for further proofreading. After proofreading, corrections have been made accordingly throughout the document.
Reviewer 3 Report
I would suggest tight further junction analysis would required.
Author Response
We have again included the Tjp1 -figure and results in the manuscript. We agree with the reviewer that more markers could be included. Unfortunately, due to the ongoing Covid-19 pandemic, we are currently unable to perform complementary histological analyses at the given time frame, which is 5 days. Currently, the situation is very bad in Finland, especially where we live and all our laboratories are closed. Further, at the university we are only allowed to do remote work, which means that the only corrections that we can do the manuscript now are the ones that can be done with the computer. We are very sorry about this inconvenience and we hope that the reviewer understands this situation. During the first round of revision, we were able to provide the requested results because the experiments were done before the Covid-19 pandemic started.
But definitely, those are worth of publishing in a future work whenever it will be possible to work in the laboratory.
Therefore, we have added to the discussion lines 693-698, page 24 the following:
However, XOS did not affect intestinal Tjp1 according to our immunohistological analysis. The latter finding was somewhat surprising because we have earlier shown that F. prausnitzii administration increased the expression of intestinal Tjp1 mRNA [23], and others have shown that prebiotic oligosaccharides in general can prevent HFD-induced impairment of gut permeability [39]. Thus, further studies are needed to determine the effects of XOS on other tight junction proteins.